# DiffuSpec: Unlocking Diffusion Language Models for Speculative Decoding

## Abstract

As large language models (LLMs) scale up, accuracy improves, but the autoregressive (AR) nature of decoding increases latency, since each token requires a serial forward pass. Speculative decoding addresses this by employing a fast drafter to propose multi-token drafts, which are then verified in parallel by the target model. However, many deployments still rely on AR drafters, whose sequential passes limit wall-clock gains. We revisit the drafting stage and present **DiffuSpec**, a speculative decoding scheme that reuses an existing pretrained diffusion language model (DLM) as a drafter to produce multi-token drafts in a single forward pass, while remaining compatible with standard AR verifiers. Because DLM drafts are generated under bidirectional conditioning, parallel per-position candidates form a token lattice in which the locally highest-probability token at each position need not form a causal left-to-right path. Moreover, DLM drafting requires pre-specifying a draft length, inducing a speed–quality trade-off. To address these challenges, we introduce two practical components: (i) a causal-consistency path search (CPS) over this lattice that extracts a left-to-right path aligned with AR verification; and (ii) an adaptive draft-length (ADL) controller that adjusts the length of the next proposal based on recent acceptance feedback and realized generated length. Across benchmarks, DiffuSpec yields up to $3\times$ wall-clock speedup and outperforms strong baselines, showing that diffusion-based drafting can be a competitive alternative to autoregressive drafters for speculative decoding.

## 1 Introduction

Large language models (LLMs) continue to improve with scale, yet autoregressive (AR) decoding remains a latency bottleneck because generating $K$ tokens requires $K$ serial forward passes (Leviathan et al., 2023; Hoffmann et al., 2022). A common line of work accelerates inference via pruning and sparsity, quantization, or knowledge distillation, but these techniques often introduce accuracy trade-offs or additional engineering complexity (Frantar et al., 2022; Frantar & Alistarh, 2023; Xu et al., 2024). Speculative decoding offers a nearly lossless alternative: a fast drafter first proposes multi-token drafts, and then the target model verifies the drafts in parallel, which preserves the target distribution while reducing wall-clock time (Xia et al., 2024). However, the speedup hinges on two factors: the drafter's per-step drafting throughput and the verification acceptance rate, defined as the fraction of drafted tokens accepted by the AR verifier during parallel verification.

In practice, existing speculative decoding methods primarily try to improve one or both of these factors. One line of work focuses on faster draft generation by using a smaller autoregressive drafter (Leviathan et al., 2023; Chen et al., 2023) (Fig. 1a) or retrieval-based proposals (He et al., 2023; Saxena, 2023), but the acceptance rate can be limited when the drafter does not approximate the target distribution well. A complementary line, exemplified by EAGLE-style methods (Li et al., 2024a;b; 2025), explicitly trains or calibrates the drafter to better match the target model, which substantially improves acceptance. However, the drafter itself remains sequential and still requires one forward pass per drafted token, so the overall throughput is fundamentally bounded by autoregressive generation. Even multi-token prediction (MTP) schemes such as Medusa (Cai et al., 2024), which attach auxiliary heads to predict several future tokens in parallel from the current AR state, ultimately advance the sequence one MTP step at a time; the maximum effective draft length per step is bounded by the number and depth of heads, so end-to-end acceleration remains constrained by the underlying autoregressive backbone.

Recent advances in diffusion language models (DLMs) (Li et al., 2022) open a new avenue for speculative decoding. Several pretrained DLMs (Fig. 1b), such as Dream-style models, are obtained by fine-tuning autoregressive LMs and therefore remain well aligned with the token distribution of larger AR targets in the same family, while at the same time being able to propose a block of token candidates in a single forward pass (Ye et al., 2025). These capabilities directly match drafter desiderata—higher per-step drafting throughput and strong proposal quality—making DLMs a compelling fit for parallel generation with parallel verification. However, DLM proposals are generated under bidirectional conditioning rather than strict left-to-right causality. This induces a diffusion token lattice over per-position candidates, where the locally highest-probability token at each position need not define a causal left-to-right path. In addition, DLM drafting requires specifying a draft length in advance. Together, these properties raise two practical questions we study: (i) **causal alignment**: how to select, from this lattice, a left-to-right path aligned with AR verification to maximize acceptance; and (ii) **draft length**: how to choose the block size to balance drafting cost against verification acceptance, since longer drafts increase proposal cost without guaranteeing higher acceptance. While prior work such as SpecDiff (Christopher et al., 2024) has begun to explore diffusion-based drafters, a systematic treatment of causal alignment and draft-length selection in this setting remains under-explored.

To address these two issues, we present **DiffuSpec**, which reuses a pretrained DLM as the drafter in place of the usual autoregressive model and wraps it with two lightweight components: (i) a *causal-consistency path search* (CPS) over the diffusion token lattice that selects a left-to-right path aligned with AR verification to improve acceptance; and (ii) an *adaptive draft-length* (ADL) controller that sets the next draft length based on recent acceptance statistics and the realized generated length. DiffuSpec is implemented as a plug-in drafter module on top of an SPS-style speculative decoding interface (Leviathan et al., 2023): it only replaces the drafter side of the standard drafter–verifier interface, requires no architectural changes to the target model, and integrates into existing serving stacks with minimal modification. Across diverse generation tasks, DiffuSpec delivers up to $3\times$ wall-clock speedup over strong baselines.

In summary, our main contributions include:

- We introduce pretrained DLMs as drafters for speculative decoding and analyze two defining traits—bidirectional conditioning and preset draft length—showing how they jointly affect verifier acceptance and end-to-end speedup and what challenges they pose.

- We propose **DiffuSpec**, a speculative decoding scheme that reuses a pretrained DLM as a drafter and adds two lightweight components: CPS, which extracts a causal path from the diffusion token lattice, and ADL, which sets the next draft length from recent acceptance statistics and generated length; DiffuSpec plugs into an SPS-style interface and works with existing AR verifiers with minimal serving-stack changes.

- We demonstrate that DiffuSpec achieves up to $3\times$ wall-clock speedup across tasks, outperforming strong speculative decoding baselines and showing that DLMs are effective drafters for speculative decoding.

## 2 RELATED WORK

**Speculative decoding.** Speculative decoding accelerates autoregressive (AR) generation by letting a fast *drafter* propose multiple tokens that a target LM verifies in parallel, while preserving the target distribution (Xia et al., 2024; Sun et al., 2025). Existing methods differ mainly in how the drafter is obtained and how verification is organized. One line of work uses a smaller pretrained AR drafter (Leviathan et al., 2023; Chen et al., 2023) or *retrieval/cache*-based drafters that mine recent $n$-grams or suffix structures (He et al., 2023; Saxena, 2023), often combined with verification-side improvements such as block verification and massively parallel cache-tree validation (Sun et al., 2024; Miao et al., 2024; Svirschevski et al., 2024). Another line reduces strict step-by-step dependency without an auxiliary drafter via *lookahead* updates (Fu et al., 2024). A third line attaches multi-token prediction (MTP) heads to the target LM (Cai et al., 2024; Ankner et al., 2024) or distills a separate drafter that operates at the feature/token level (Li et al., 2024a;b; 2025). However, the first two lines often suffer from limited acceptance rates under distribution mismatch or weak

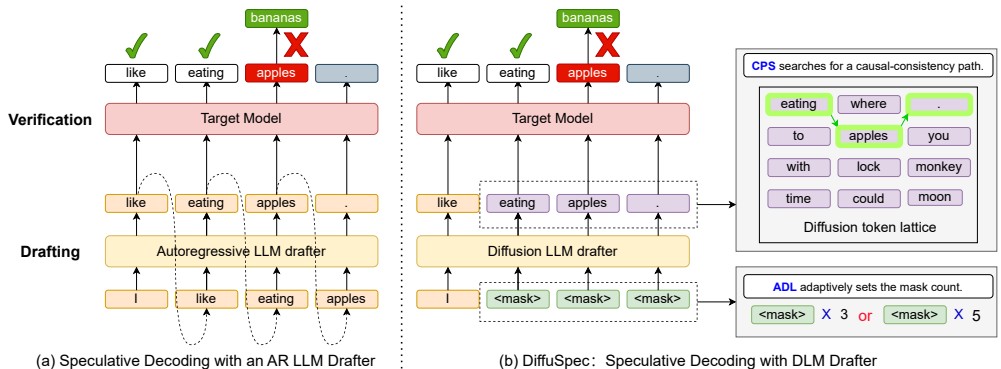

Figure 1: **Speculative decoding: AR vs. DiffuSpec.** (a) **AR drafter:** drafts are produced sequentially and then block-verified by the target AR model. (b) **DiffuSpec (DLM drafter):** a single forward pass proposes a block for one-shot parallel verification; within DiffuSpec, causal-consistency path search (CPS) selects a left-to-right path from the diffusion token lattice, and the adaptive draft-length (ADL) controller sets the next draft length by selecting how many masked positions to fill.

retrieval, while methods in the third line require additional training and interface changes, with their drafters remaining fundamentally autoregressive and incurring non-negligible drafting latency.

**Diffusion language models.** Discrete/latent diffusion for text ranges from early D3PMs (Austin et al., 2021) and Diffusion-LM (Li et al., 2022) to hybrids with pretrained LMs (Zhou et al., 2023; He et al., 2022) and recent scaling/adaptation frameworks (Gong et al., 2024). Large pretrained DLMs have been reported to be competitive with similarly sized AR baselines while retaining diffusion-style parallel refinement (Nie et al., 2025; Ye et al., 2025). In particular, Dream-like models (Ye et al., 2025) are finetuned from strong AR LMs in the same family, so their token distributions remain well aligned with larger AR targets, making them especially suitable as drafters for speculative decoding. At inference time, DLMs natively support parallel multi-token updates with iterative refinement but pay for bidirectional attention and multiple denoising steps; this has motivated deployment-time accelerators such as adaptive KV caching, dynamic cache eviction, and suffix-dropout pruning (Liu et al., 2025; Song et al., 2025; Chen et al., 2025). These traits—single- or few-pass proposal of token blocks together with strong proposal quality—make DLMs promising candidates as drafters for speculative decoding.

**Diffusion as a drafter for speculative decoding.** Christopher et al. (2024) first showed that a discrete diffusion model can draft sequences for AR verification, validating the feasibility of diffusion-based drafting. However, prior work typically (i) trains or calibrates a dedicated diffusion drafter and (ii) lacks a systematic analysis of how draft length and the diffusion-induced token lattice with relaxed causality interact with AR verification. In contrast, **DiffuSpec** reuses pretrained DLMs as drafters and introduces (a) a *causal-consistency path search* (CPS) over the diffusion-induced token lattice and (b) an *adaptive draft-length* (ADL) controller that together improve accepted prefixes and wall-clock speedups under AR block verification.

## 3 PRELIMINARIES—SPECULATIVE DECODING

Let $p_\theta$ be the target autoregressive (AR) language model and $\mathbf{x}_{1:j}$ the current prefix. Speculative decoding (Leviathan et al., 2023; Chen et al., 2023; Xia et al., 2024) accelerates generation under a *drafter–verifier* interface: a fast drafter proposes a short continuation, and the target AR model verifies it in parallel while preserving the $p_\theta$ distribution.

**Drafting.** Given $\mathbf{x}_{1:j}$, a drafter $q_\phi$ proposes a length-$k_t$ block $\hat{\mathbf{y}}_{j+1:j+k_t} = (\hat{y}_{j+1}, \ldots, \hat{y}_{j+k_t})$ conditioned on $\mathbf{x}_{1:j}$, and records per-position conditional probabilities $\{q_\phi(\hat{y}_{j+i} \mid \mathbf{x}_{1:j+i-1})\}_{i=1}^{k_t}$. Here $t = 1, 2, \ldots$ indexes speculative steps.

**Parallel verification.** The target model evaluates the drafted tokens in a single parallel pass, producing $\{p_\theta(\hat{y}_{j+i} \mid \mathbf{x}_{1:j+i-1})\}_{i=1}^{k_t}$, and then processes them left-to-right with the standard acceptance

rule:

$$\alpha_{t,i} = \min\left(1, \frac{p_\theta(\hat{y}_{j+i} \mid \mathbf{x}_{1:j+i-1})}{q_\phi(\hat{y}_{j+i} \mid \mathbf{x}_{1:j+i-1})}\right), \quad i = 1, \ldots, k_t. \quad (1)$$

If $\hat{y}_{j+i}$ is rejected, a replacement is sampled from the residual distribution proportional to $\left[p_\theta(\cdot \mid \mathbf{x}_{1:j+i-1}) - q_\phi(\cdot \mid \mathbf{x}_{1:j+i-1})\right]_+$, where $[u]_+ = \max(u, 0)$, followed by normalization; all remaining drafted tokens are discarded before continuing. This procedure is unbiased with respect to $p_\theta$ (Leviathan et al., 2023, App. A.1) and admits verifier-side engineering such as block or tree-based parallel verification to further reduce latency (Sun et al., 2024; Miao et al., 2024).

**Accepted prefix length.** At speculative step $t$ with proposal length $k_t$, let $A_{t,i} \in \{0, 1\}$ indicate whether the $i$-th drafted token is accepted by the verifier *given* that positions $1{:}i-1$ were accepted. The number of tokens actually committed is

$$L_t^{\text{acc}} = \max\{m \in \{0, \ldots, k_t\} : A_{t,1} = \cdots = A_{t,m} = 1\} = \sum_{i=1}^{k_t} \prod_{r=1}^{i} A_{t,r}. \quad (2)$$

The verifier appends the accepted prefix $\hat{\mathbf{y}}_{j+1:j+L_t^{\text{acc}}}$ and discards the remainder, yielding the updated prefix $\mathbf{x}_{1:j+L_t^{\text{acc}}}$. Decoding terminates early if an EOS token is accepted. We use $L_t^{\text{acc}}$ as a per-step measure of useful progress; holding latency fixed, larger values imply higher speedup.

## 4 DIFFUSPEC

As shown in Fig. 1b, **DiffuSpec** departs from conventional speculative decoding by replacing the AR drafter with a pretrained diffusion language model (DLM) that proposes a length-$k_t$ draft in a single forward pass, and by augmenting drafting with *causal-consistency path search* (CPS) and an *adaptive draft-length* (ADL) controller. We next describe these three components in turn.

### 4.1 DLM AS A TRAINING-FREE DRAFTER

Unlike autoregressive models with fixed left-to-right factorization, diffusion language models learn a non-autoregressive denoising mapping that reconstructs clean text from corrupted text (Austin et al., 2021; Gong et al., 2024; Nie et al., 2025; Ye et al., 2025; Chen et al., 2025).

**Training.** Let $\mathbf{x}^{(0)}$ be a clean sequence and $\mathbf{x}^{(\eta)}$ its corrupted counterpart at noise level $\eta \in [0, 1]$. We define a forward corruption kernel $r$ with a user-specified discrete noise prior $\pi_{\text{noise}}$:

$$r\left(x_i^{(\eta)} \mid x_i^{(0)}\right) = (1 - \eta)\,\mathbf{1}\{x_i^{(\eta)} = x_i^{(0)}\} + \eta\,\pi_{\text{noise}}\left(x_i^{(\eta)}\right), \quad (3)$$

where $\sum_v \pi_{\text{noise}}(v) = 1$ (e.g., all mass on [MASK] or a mixture over noise symbols). A parameterized denoiser $q_\phi$ is trained with token-wise cross-entropy to predict originals at corrupted positions:

$$\mathcal{L}(\phi) = -\mathbb{E}_{\eta, \mathbf{x}^{(0)}, \mathbf{x}^{(\eta)}}\left[\sum_{i:\,x_i^{(\eta)} \neq x_i^{(0)}} \log q_\phi\left(x_i^{(0)} \mid \mathbf{x}^{(\eta)}\right)\right], \quad (4)$$

where $q_\phi$ is a Transformer with bidirectional attention.

**Inference (iterative refinement).** Given a prefix $\mathbf{r} = \mathbf{x}_{1:j}$ and target length $k_t$, initialize $\mathbf{y}^{(0)} = \mathbf{r} \circ ([\text{MASK}])^{k_t}$ with masked set $M_0 = \{j+1, \ldots, j+k_t\}$, where $\circ$ denotes concatenation. For refinement steps $s = 1, \ldots, S$, compute per-position conditionals $q_\phi(y_i \mid \mathbf{y}^{(s-1)})$ for $i \in M_{s-1}$, choose an update subset $U_s \subseteq M_{s-1}$ (e.g., top-$K$ by confidence), and set

$$y_i^{(s)} = \begin{cases} \arg\max_{v \in \mathcal{V}} q_\phi(y_i = v \mid \mathbf{y}^{(s-1)}), & i \in U_s, \\ y_i^{(s-1)}, & \text{otherwise,} \end{cases} \quad M_s = M_{s-1} \setminus U_s, \quad (5)$$

until $M_s = \varnothing$. By default we use a single refinement pass ($S{=}1$) to isolate drafting cost; $S{>}1$ is ablated in Sec. 5.

**Integration with speculative decoding.** At speculative step $t$ with prefix $\mathbf{x}_{1:j}$, a pretrained DLM proposes a length-$k_t$ block $\hat{\mathbf{y}}_{j+1:j+k_t}$ in essentially one orward/refinement pass, and can expose

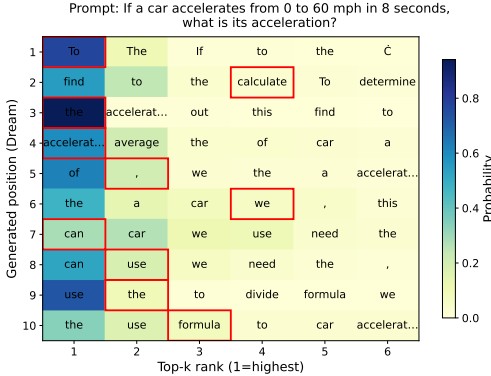

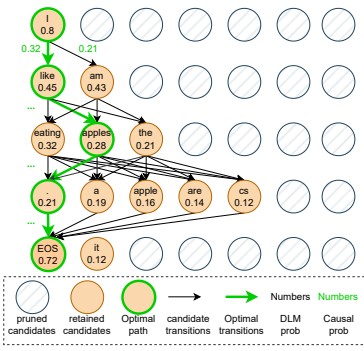

Figure 2: **DLM token-mass diffusion (Dream-7B).** Probability mass spreads across positions during joint block refinement; the per-position top-1 need not yield an AR-consistent left-to-right path under $p_\theta$.

Figure 3: **Pruned candidate lattice and CPS.** We keep tokens via a cumulative-mass threshold $\tau$ (e.g., $0.8$), always retain EOS, early-stop after the first EOS, and select the best path using a DLM score plus a causal ($n$-gram) proxy.

per-position top-$M$ candidate sets with log-scores taken under the current draft context $\mathbf{y}^{(S)}$. For verifier-side acceptance with a DLM drafter, we evaluate a left-to-right proxy by masking all future draft positions when scoring the token at $j{+}i$:

$$q_\phi^{\text{L2R}}(v \mid \mathbf{x}_{1:j+i-1}) := q_\phi\Big(v \mid \mathbf{x}_{1:j} \circ \underbrace{\big([\text{MASK}]\big)^{i-1}}_{\text{past in-block}}, \underbrace{\big([\text{MASK}]\big)^{k_t-i+1}}_{\text{future in-block}}\Big). \tag{6}$$

We use $q_\phi^{\text{L2R}}$ in the standard acceptance ratio. Accordingly, Sec. 4.2 introduces **CPS** to align proposals causally with the AR verifier, and Sec. 4.3 presents **ADL** to set $k_t$ near the speed–quality sweet spot.

### 4.2 CAUSAL-CONSISTENCY PATH SEARCH (CPS)

**Phenomenon and motivation.** Under relaxed causality, the DLM refines tokens jointly within a block. As a result, token probability mass spreads across positions and the per-position top-1 chosen by the DLM is not necessarily the best left-to-right choice for the AR verifier $p_\theta$ (Fig. 2). To mitigate this mismatch, we explicitly search before verification—for a left-to-right path that is both high-confidence under the DLM and fluent under a causal proxy (Fig. 3).

**Lattice and pruning.** We first specify the search space. From the final DLM pass, for each position $i = 1{:}k_t$ we extract a candidate set $\mathcal{C}_{j+i}$ (top-$M$) with log-scores $\ell_{j+i}^{\text{dlm}}(v) = \log q_\phi\big(v \mid \mathbf{x}_{1:j}, \mathbf{y}_{\setminus(j+i)}^{(S)}\big)$, i.e., conditioning on the current draft context except the target position. The naive Cartesian product over $\{\mathcal{C}_{j+i}\}_{i=1}^{k_t}$ is exponential, so we apply a training-free, mass-adaptive pruning rule that respects local uncertainty. Let $p_{j+i}(v) = \exp(\ell_{j+i}^{\text{dlm}}(v))$. We retain the smallest prefix exceeding a cumulative-mass threshold $\tau$:

$$M_i = \min\Big\{m \le M_{\max} : \sum_{v \in \text{Top-}m} p_{j+i}(v) \ge \tau\Big\}, \qquad \mathcal{C}_{j+i} \leftarrow \text{Top-}M_i. \tag{7}$$

This makes $|\mathcal{C}_{j+i}|$ entropy-adaptive—peaky positions keep few candidates; flatter ones keep more, capped by $M_{\max}$. In addition, we stop expanding once the first EOS is placed: diffusion proposals tend to pad with EOS after the content is "complete" (qualitative trend in Fig. 4), so exploring beyond the first EOS rarely yields causal gains.

**Scoring and search.** Let $m_{\max}$ denote the depth up to (and including) the first EOS encountered during expansion. Given the pruned lattice, we score $\pi = (\pi_1, \dots, \pi_m)$ by combining DLM confidence with a small causal proxy (e.g., an $n$-gram or a tiny causal LM):

$$\mathcal{S}(\pi) = \sum_{i=1}^{m} \Big[\lambda\, \ell_{j+i}^{\text{dlm}}(\pi_i) + (1-\lambda)\, \ell_{j+i}^{\text{ng}}(\pi_{1:i})\Big], \tag{8}$$

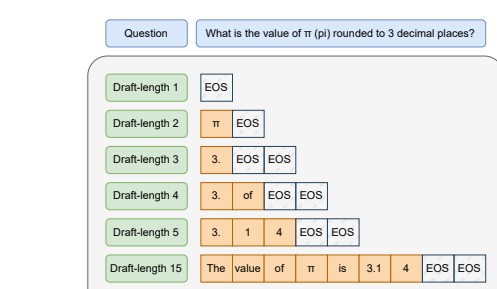

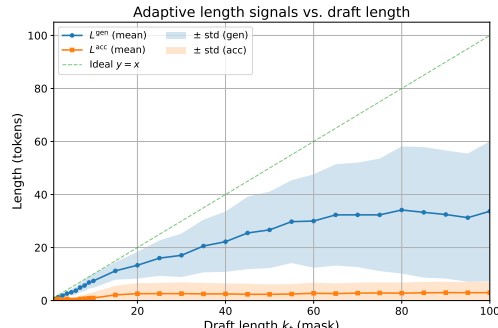

Figure 4: **Qualitative effect of draft length.** As the draft length $k_t$ increases, DLM proposals evolve from short fragments to more complete answers; once the model deems the content "complete," an early EOS truncates further content.

Figure 5: **Adaptive-length signals vs. draft length.** For each $k_t$, we plot the mean and $\pm 1$ standard deviation of the EOS-aware generation length $L^{\text{gen}}$ and the accepted length $L^{\text{acc}}$ across evaluation prompts. The dashed diagonal $y=x$ marks the ideal should-generate line.

where $\ell^{\text{ng}}_{j+i}$ is the causal proxy log-score of $\mathbf{x}_{1:j} \circ \pi_{1:i}$ and $\lambda \in [0,1]$ trades off between DLM confidence and causal fluency. We then run left-to-right beam search (beam $B$) on the pruned lattice until EOS is placed. If $\bar{C}$ denotes the average branching factor after pruning, the per-step complexity is $O(B\,\bar{C}\,m_{\max})$. As $\tau \to 1$ and $B$ increases, the result approaches the unpruned optimum.result approaches the unpruned optimum.

**Effect.** By entropy-adaptive pruning, early stopping at the first EOS, and the causal–denoising score in equation 8, CPS pushes the first $p_\theta - q_\phi$ mismatch farther to the right, thereby increasing the expected accepted length $L^{\text{acc}}_t$ and improving end-to-end speed.

### 4.3 ADAPTIVE DRAFT LENGTH (ADL)

**Phenomenon and motivation.** Draft length $k_t$ jointly determines drafting cost, proposal quality, and verifier acceptance. Short drafts often yield terse fragments; moderate drafts capture more complete reasoning; very long drafts saturate content and trigger early EOS while also accumulating off-path tokens that the verifier rejects (Fig. 4). Empirically, the EOS-aware generation length $L^{\text{gen}}$ increases with $k_t$ and then saturates, and the accepted length $L^{\text{acc}}$ tracks it (Fig. 5). The saturation point, however, is instance dependent and varies across prompts and along the trajectory, leading to large variance as shown in Fig. 5. A fixed $k_t$ therefore either wastes compute when too long or throttles progress when too short, which motivates an adaptive controller.

**Signals.** Given the drafted block $\hat{\mathbf{y}}_{j+1:j+k_t}$, let $s_t$ be the index of the first EOS (or $+\infty$ if none) and define the EOS-aware generation signal

$$L^{\text{gen}}_t = \min(s_t - 1, \ k_t). \tag{9}$$

We compute $s_t$ from the raw DLM draft before CPS; since CPS also early-stops at the first EOS, both signals are aligned. After parallel verification we obtain the accepted prefix length $L^{\text{acc}}_t$ as defined in Sec. 3. To reduce volatility from occasional early EOS or transient rejections, we use exponential moving averages:

$$\tilde{L}^{\text{gen}}_t = (1-\rho)\tilde{L}^{\text{gen}}_{t-1} + \rho L^{\text{gen}}_t, \qquad \tilde{L}^{\text{acc}}_t = (1-\rho)\tilde{L}^{\text{acc}}_{t-1} + \rho L^{\text{acc}}_t, \qquad \rho \in (0,1]. \tag{10}$$

**Controller.** With guardrails $k_{\min} \le k_{t+1} \le k_{\max}$, we adopt a one-line $O(1)$ policy:

$$k_{t+1} \ = \ \text{clip}\Big( \lceil \tilde{L}^{\text{gen}}_t + \delta\,\mathbf{1}\{\tilde{L}^{\text{acc}}_t \ge \tilde{L}^{\text{gen}}_t\} \rceil, \ k_{\min}, \ k_{\max} \Big), \tag{11}$$

where $\text{clip}(z,a,b) = \min\{\max\{z,a\},b\}$ and $\delta > 0$ is a small growth increment that activates when the verifier keeps up, namely when the accepted length matches the generated length on average. Intuitively, $\tilde{L}^{\text{gen}}_t$ estimates how much content the DLM is ready to produce before EOS, and $\tilde{L}^{\text{acc}}_t$

---

**Algorithm 1:** DIFFUSPEC (4-stage): DLM drafting + CPS + parallel verification + ADL

---

**Input:** prefix $\mathbf{x}_{1:j}$; target LM $p_\theta$; DLM $q_\phi$; ADL params $(k_{\min}, k_{\max}, \delta, \rho)$; CPS params
$\quad\quad (M_{\max}, \tau, B, \lambda)$.
**Init:** $\tilde{L}_0^{\text{gen}} \leftarrow 0$, $\tilde{L}_0^{\text{acc}} \leftarrow 0$; set $k_1 \leftarrow k_{\max}$.
**for** $t = 1, 2, \ldots$ *until termination* **do**
    **(1) Draft:** run DLM to produce a length-$k_t$ block and per-position candidate sets
    $\{\mathcal{C}_{j+i}\}_{i=1}^{k_t}$ (top-$M_{\max}$ ) with scores $\ell_{j+i}^{\text{dlm}}(\cdot)$.
    **(2) CPS:** on a pruned candidate lattice (cumulative-mass $\tau$, always keep EOS, early-stop
    after the first EOS), run left-to-right beam search (beam $B$) using score $\mathcal{S}(\cdot)$ in equation 8
    to obtain a left-to-right path $\hat{\mathbf{y}}_{j+1:j+m_t}$ *(path length $m_t$).*
    **(3) Parallel verification:** block verification of $\hat{\mathbf{y}}_{j+1:j+m_t}$ with $p_\theta$; compute acceptance
    using $q_\phi^{\text{L2R}}$; obtain $L_t^{\text{acc}}$; append the accepted prefix and update $j \leftarrow j + L_t^{\text{acc}}$; if an EOS is
    accepted, **terminate**.
    **(4) ADL:** compute $L_t^{\text{gen}}$ from the proposal's first-EOS index $s_t$; update EMAs $\tilde{L}_t^{\text{gen}}, \tilde{L}_t^{\text{acc}}$;
    set $k_{t+1}$ via equation 11.

---

indicates whether those tokens are reliably accepted; the policy increases $k_t$ only when both signals align.

**Effect.** ADL tracks the instance-specific speed–quality sweet spot in real time. As $k_t$ grows into the saturation regime, $L_t^{\text{gen}}$ plateaus and the controller stabilizes; when acceptance lags, the policy avoids oversizing drafts; when acceptance catches up, it expands gently via $\delta$.

### 4.4 TRAINING-FREE, SERVING-COMPATIBLE FRAMEWORK

As summarized in Fig. 1b and Alg. 1, each speculative step in DiffuSpec follows a fixed four-stage pipeline with no changes to the target model and only minimal serving-stack adjustments: *(i) Drafting* with a pretrained DLM to produce a length-$k_t$ block and per-position candidates; *(ii) CPS* on a pruned candidate lattice to select a left-to-right path aligned with AR causality; *(iii) Parallel verification* by the target $p_\theta$ (using $q_\phi^{\text{L2R}}$ in the acceptance ratio) to return the accepted prefix length $L_t^{\text{acc}}$ and advance the prefix; *(iv) ADL* to update the next draft length $k_{t+1}$ from the signal $L_t^{\text{gen}}$ and verifier feedback $L_t^{\text{acc}}$, within guardrails $[k_{\min}, k_{\max}]$. By improving the acceptance profile via CPS and right-sizing proposals via ADL, DiffuSpec increases $L_t^{\text{acc}}$ per step while keeping drafting cost near the speed–quality sweet spot. For correctness, when the verifier applies the standard speculative-decoding acceptance rule with $q_\phi^{\text{L2R}}$, the classical unbiasedness analysis w.r.t. $p_\theta$ applies.

## 5 EXPERIMENTS

**Datasets.** We follow the Spec-Bench protocol (Xia et al., 2024) and span six task families: *Multi-turn Conversation* (MT; Zheng et al., 2023), *Machine Translation* (Trans), *Summarization* (Sum; Nallapati et al., 2016), *Open-domain QA* (QA; Kwiatkowski et al., 2019), *Mathematical Reasoning* (Math; Cobbe et al., 2021), and *Retrieval-Augmented Generation* (RAG; Karpukhin et al., 2020). For additional details on the datasets, see Appendix A.

**Speed metrics.** We report (i) *Mean Accepted Tokens (MAT)*, the expected length of consecutively accepted prefixes per speculative step, averaged over all steps and examples; and (ii) *Speedup*, defined as end-to-end throughput relative to the AR-greedy baseline on the same target model and hardware. All timings are wall-clock and account for DLM drafting, CPS, ADL, and parallel verification. To ensure comparable quality (quality-locked setting), verification is performed with greedy decoding (temperature $= 0$), yielding task metrics statistically indistinguishable from AR-greedy.

**Baselines.** We compare DiffuSpec against a range of speculative decoding methods, including SPS (Leviathan et al., 2023), Lookahead (Fu et al., 2024), PLD (Saxena, 2023), Recycling (Luo et al., 2024), SAMD (Hu et al., 2024), EAGLE2 (Li et al., 2024a;b), EAGLE3 (Li et al., 2025), and SpecDiff (Christopher et al., 2024). To isolate the effect of our ADL controller, we also evaluate a Minions-style variant that is identical to DiffuSpec except that ADL is replaced by the length con-

| Method | Speedup ($\times$ vs. AR, $\uparrow$) | | | | | | Mean (MAT / Speedup) |
|---|---|---|---|---|---|---|---|
| | MT | Trans | Sum | QA | Math | RAG | |
| Lookahead | 1.37$\times$ | 1.16$\times$ | 1.15$\times$ | 1.33$\times$ | 1.52$\times$ | 1.21$\times$ | 1.82 / 1.30$\times$ |
| PLD | 1.83$\times$ | 1.29$\times$ | 2.76$\times$ | 1.87$\times$ | 1.55$\times$ | 2.37$\times$ | 2.11 / 1.93$\times$ |
| Recycling | 2.15$\times$ | 1.85$\times$ | 2.03$\times$ | 2.06$\times$ | 2.45$\times$ | 1.83$\times$ | 3.13 / 2.07$\times$ |
| SAMD | 1.99$\times$ | 1.54$\times$ | **3.38**$\times$ | 2.44$\times$ | 1.63$\times$ | **3.27**$\times$ | 2.18 / 2.35$\times$ |
| EAGLE2 | 2.47$\times$ | 1.56$\times$ | 1.64$\times$ | 1.91$\times$ | 3.18$\times$ | 1.64$\times$ | 3.47 / 2.09$\times$ |
| EAGLE3 | 3.01$\times$ | 2.35$\times$ | 3.03$\times$ | 2.41$\times$ | 3.12$\times$ | 2.86$\times$ | 4.28 / 2.80$\times$ |
| SPS | 1.69$\times$ | 1.64$\times$ | 1.74$\times$ | 1.50$\times$ | 1.86$\times$ | 1.62$\times$ | 6.18 / 1.67$\times$ |
| SpecDiff | 2.65$\times$ | 2.61$\times$ | 1.96$\times$ | 2.41$\times$ | 2.95$\times$ | 2.02$\times$ | 6.05 / 2.69$\times$ |
| Minions | 3.02$\times$ | 3.18$\times$ | 2.37$\times$ | 2.93$\times$ | 3.91$\times$ | 2.29$\times$ | 6.44 / 2.97$\times$ |
| **DiffuSpec** | **3.09**$\times$ | **3.38**$\times$ | 2.41$\times$ | **3.03**$\times$ | **4.02**$\times$ | 2.38$\times$ | **6.99 / 3.08**$\times$ |

Table 1: **Main results on Spec-Bench.** Per-task columns report *Speedup* only (unitless ratio vs. AR, $\uparrow$); the rightmost column reports the task-macro *Mean (MAT / Speedup)*.

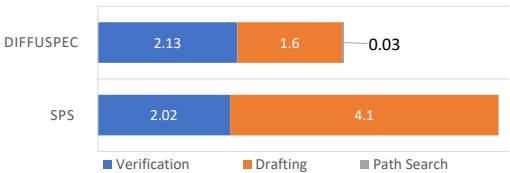

| CPS | ADL | Mean-MAT | Mean-Speedup |
|---|---|---|---|
| ✓ | ✓ | **6.99** | **3.08**$\times$ |
| ✓ | ✗ | 6.95 | 2.98$\times$ |
| ✗ | ✓ | 6.43 | 2.73$\times$ |
| ✗ | ✗ | 6.05 | 2.69$\times$ |

Figure 6: **Per-step wall-clock time (s).** Mean seconds per drafter–verifier round spent in drafting, verification, and CPS (SPS vs. DiffuSpec).

Table 2: **Ablation on DiffuSpec components.** ✓ indicates the component is enabled. Both ADL and CPS improve performance, with CPS contributing the larger share of gains.

troller from Minions (Wang et al., 2024); we refer to this variant as *Minions*. All methods are run in our unified evaluation stack under the same hardware and decoding configuration.

**Targets and drafters.** Unless otherwise stated, the target AR model $p_\theta$ is Qwen2.5-32B (Xu et al., 2025) for all methods, including ours. DiffuSpec uses Dream-7B (Ye et al., 2025) as the diffusion drafter; SpecDiff and Minions are also evaluated with Dream-7B for a fair comparison. For SPS, we follow the standard Qwen2.5-7B autoregressive drafter. Lookahead, PLD, Recycling, and SAMD do not employ a separate drafter and operate directly on the target model or its cache/retrieval stack. EAGLE2 and EAGLE3 attach trained drafter heads to Qwen2.5-32B; we implement them following their official repositories.

**Implementation details.** Experiments run on a single NVIDIA A100 (80GB) with 11 CPU cores and 100GB RAM, PyTorch 2.6.0. Following Kou et al. (2024); Luo et al. (2024), verification uses greedy decoding with batch size = 1, KV cache enabled. Unless stated, DiffuSpec uses a single diffusion refinement step ($S=1$) to isolate drafting cost. Controller and search hyperparameters are fixed across tasks: $k_{\min}=20$, $k_{\max}=30$, beam size $B=3$, mass threshold $\tau=0.8$, per-position cap $M_{\max}=15$, mixing weight $\lambda=0.5$, controller increment $\delta=10$ tokens, and EMA smoothing $\rho=0.5$. The causal proxy is a 3-gram KenLM trained offline on a text corpus and reused across all tasks.

## 5.1 EFFECTIVENESS

**Overall comparison.** Tab. 1 summarizes wall-clock speedups on Spec-Bench. DiffuSpec achieves the best overall performance, with a Mean-MAT of 6.99 and a Mean-Speedup of 3.08$\times$, outperforming all speculative decoding baselines under the same Qwen2.5-32B target, hardware, and decoding configuration. Compared to strong autoregressive competitors, DiffuSpec improves both quality and efficiency (e.g., vs. SPS: +0.81 MAT and +1.41$\times$ speedup; vs. EAGLE3: +2.71 MAT and +0.28$\times$ speedup), even though EAGLE3 requires additional training and architectural modifications. At the task level, DiffuSpec attains the highest speedups on *MT/Trans/QA/Math*, with 3.09$\times$/3.38$\times$/3.03$\times$/4.02$\times$, indicating consistently longer accepted prefixes and faster end-to-end progress at matched quality.

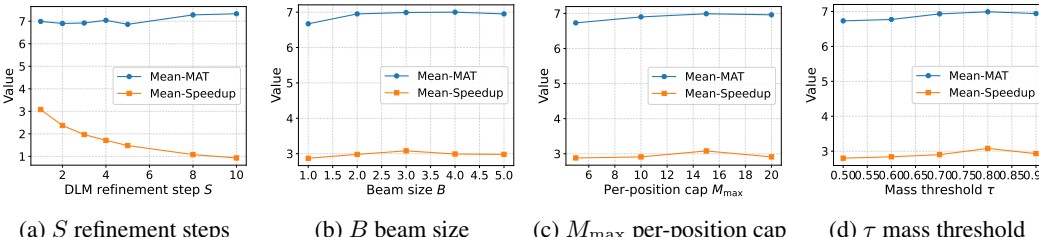

(a) $S$ refinement steps    (b) $B$ beam size    (c) $M_{\max}$ per-position cap    (d) $\tau$ mass threshold

Figure 7: **Sensitivity to decoding/search hyperparameters.** Each panel plots *Mean-MAT* and *Mean-Speedup* versus a single knob under the quality-locked setting.

**Comparison to diffusion-based and adaptive-length baselines.** Among diffusion-based methods, SpecDiff already demonstrates that a DLM can serve as an effective drafter, but DiffuSpec further improves both metrics under an identical Qwen2.5-32B + Dream-7B configuration (Mean-MAT: 6.99 vs. 6.05; Mean-Speedup: $3.08\times$ vs. $2.69\times$). This gain aligns with our design goals: CPS explicitly extracts a causal left-to-right path from the diffusion-induced token lattice, while ADL adapts the draft length toward the speed–quality sweet spot. Replacing ADL with a Minions-style acceptance-only controller yields the "Minions" row in Tab. 1; DiffuSpec still achieves higher Mean-MAT (6.99 vs. 6.44) and Mean-Speedup ($3.08\times$ vs. $2.97\times$), with consistent improvements across all task families (e.g., $3.09\times$ vs. $3.02\times$ on MT and $4.02\times$ vs. $3.91\times$ on Math).

**Where the speedup comes from.** Fig. 6 decomposes wall-clock time into *drafting*, *verification*, and *CPS search*. DiffuSpec reduces drafting cost relative to SPS by using a single DLM forward pass to propose multiple tokens, while CPS adds only minor overhead (averaging $1.1\%$ across tasks). In our setup, SPS employs a 7B AR drafter close to the target's capacity; the resulting sequential passes dominate wall-clock and blunt the benefits of verifier parallelism—MAT remains relatively high, yet end-to-end speedup is modest. By contrast, DiffuSpec with Dream-7B achieves substantially larger speedups at comparable or higher MAT by combining two levers: (i) higher per-step drafting throughput (non-AR DLM pass) and (ii) higher acceptance via *CPS*, with *ADL* right-sizing proposals. Crucially, latency is governed by the *number of drafter and target passes per accepted token*, not the drafter's parameter count alone: although a single Dream-7B pass is heavier than a single 1-layer drafter pass, its higher MAT and parallel drafting allow DiffuSpec to commit a similar number of tokens with far fewer total passes, resulting in lower end-to-end wall-clock time on the same Qwen2.5-32B target. Together, these mechanisms translate acceptance gains into tangible wall-clock acceleration.

### 5.2 ABLATION

Tab. 2 quantifies the contributions of CPS and ADL. In the *no-CPS* variant, we skip causal-consistency path search and directly form the draft by taking, at each position, the highest-probability token from the DLM marginal (argmax) for a given draft length, which is then passed to the standard SPS-style block verifier. In the *no-ADL* variant, we disable the controller and fix the draft length to a constant for all steps, so that CPS operates on a token lattice of fixed width. Enabling either module improves both *Mean-MAT* and *Mean-Speedup* over the plain variant, while enabling both yields the best overall performance (6.99 MAT, $3.08\times$). Compared to the plain system (6.05 / $2.69\times$), *CPS-only* raises MAT by $+0.90$ and speedup by $+0.29\times$, whereas *ADL-only* adds $+0.38$ MAT and $+0.04\times$, respectively. Thus, CPS accounts for most acceptance gains—consistent with its role in aligning diffusion proposals with AR causality—while ADL primarily translates these gains into wall-clock speedup by adaptively setting $k_t$. When combined, they deliver a total improvement of $+0.39\times$ over the plain system. Additional analysis of draft-length choices is provided in Appendix C, and task-wise ablations with full results appear in Appendix B (Tab. 4).

### 5.3 HYPERPARAMETER SENSITIVITY

Across decoding/search knobs (Fig. 7a–7d), we observe consistent speed–quality trade-offs under the quality lock. Increasing the number of DLM refinement steps $S$ improves proposal quality and acceptance (Mean-MAT $6.99 \rightarrow 7.33$ from $S{=}1$ to 10; Fig. 7a), but substantially reduces

throughput (Mean-Speedup $3.08\times \rightarrow 0.93\times$), so we fix $S{=}1$. Enlarging the CPS beam $B$ improves causal paths and modestly raises Mean-MAT, peaking around $B{=}3{\sim}4$ (Fig. 7b); however, overhead causes speedup to plateau or regress beyond $B{=}3$, so we set $B{=}3$. Increasing the per-position cap $M_{\max}$ relaxes pruning and helps until $M_{\max}{\approx}15$ (Fig. 7c); further branching yields negligible gains and slightly hurts speed, motivating our choice of $M_{\max}{=}15$. Raising the mass threshold $\tau$ retains more local probability and improves acceptance/speed up to $\tau{\approx}0.8$ (Fig. 7d); higher values add compute with little benefit, so we use $\tau{=}0.8$. Overall, CPS-related knobs $(B, M_{\max}, \tau)$ are robust over a broad range, while multi-step refinement $S$ trades acceptance for latency. Orthogonally, ADL controls proposal size, helping convert CPS-driven acceptance gains into wall-clock acceleration.

## 6  CONCLUSION AND FUTURE WORK

We introduced **DiffuSpec**, a speculative decoding scheme that reuses a pretrained diffusion language model (DLM) as the drafter. To reconcile diffusion-based drafting with AR verification, we proposed *causal-consistency path search* (CPS) and an *adaptive draft-length* (ADL) controller. Across six task families, DiffuSpec produces high-quality multi-token drafts, delivering the strongest speedups among competing speculative decoding baselines and approaching training-based systems under quality-locked settings. Ablations indicate that both CPS and ADL improve acceptance and throughput: CPS yields the larger gains by aligning proposals with AR causality, whereas ADL stabilizes proposal size to avoid over-/under-drafting. DiffuSpec requires no additional neural training once a suitable DLM drafter is available and integrates with existing targets with minimal serving-stack changes. A current bottleneck is largely ecosystem-driven rather than algorithmic: in SPS-style cascades, smaller AR variants are routinely released alongside the target family, whereas diffusion LMs and their scaled checkpoints are still much less common. When no compatible DLM exists, one may need to train or adapt a drafter or rely on heterogeneous-vocabulary alignment, but a single DLM can then be reused across multiple targets and tasks, partially amortizing this cost. Looking ahead, we see opportunities for further system-level acceleration of DLM drafting, stronger proposal-selection objectives, and richer adaptive controllers that jointly tune draft length and search breadth, and we hope DiffuSpec serves as a practical blueprint for bridging diffusion-based generation with fast verifier-aligned decoding.

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

## A  DATASET AND IMPLEMENTATION DETAILS

**Datasets.**  We follow SPEC-BENCH (Xia et al., 2024) across six task families, using the official splits and preprocessing; prompts match §5. For *Multi-turn Conversation* (MT) we use **MT-Bench** with pairwise judging (Zheng et al., 2023). *Machine Translation* (Trans) follows Spec-Bench's public WMT-style news configuration. *Summarization* (Sum) is **CNN/DailyMail** (Nallapati et al., 2016). *Open-domain QA* (QA) is **Natural Questions** (Kwiatkowski et al., 2019). *Mathematical Reasoning* (Math) uses **GSM8K** (Cobbe et al., 2021). *Retrieval-Augmented Generation* (RAG) follows the **DPR** pipeline over Wikipedia (Karpukhin et al., 2020).

| Task | Dataset(s) | Metric(s) |
|------|------------|-----------|
| MT | MT-Bench (Zheng et al., 2023) | Win rate (pairwise) |
| Trans | Spec-Bench translation (WMT-style) | BLEU |
| Sum | CNN/DailyMail (Nallapati et al., 2016) | ROUGE-L |
| QA | Natural Questions (Kwiatkowski et al., 2019) | EM / F1 |
| Math | GSM8K (Cobbe et al., 2021) | Accuracy |
| RAG | DPR over Wikipedia (Karpukhin et al., 2020) | Accuracy |

Table 3: Spec-Bench datasets and evaluation metrics used in our experiments.

**Implementation details.**  We build our evaluation harness on top of SPEC-BENCH (Xia et al., 2024), reusing its official data loaders, prompt templates and stop criteria. All systems share the same hardware/software stack as §5 (single NVIDIA A100 80GB, 11 CPU cores, 100GB RAM, PyTorch 2.6.0). Verification uses greedy decoding (temperature $= 0$) with batch size $= 1$ and KV cache enabled; we report tokens/s averaged over the full evaluation set, excluding model-loading and first-batch warmup. Wall-clock timing includes tokenization, drafter forward(s), path search, verifier forward, and residual sampling.

Unless otherwise stated, DiffuSpec adopts a single diffusion refinement step ($S{=}1$). Controller and search hyperparameters are fixed across tasks: $k_{min}{=}20$, $k_{max}{=}30$, beam size $B{=}3$, mass threshold $\tau{=}0.8$, per-position cap $M_{max}{=}15$, mixing weight $\lambda{=}0.5$, controller increment $\delta{=}10$, and EMA smoothing $\rho{=}0.5$. The causal proxy is a 3-gram KenLM fitted *only* on the training split of each dataset (no test leakage). Speedup is defined as a unitless ratio: throughput(method) / throughput(AR-greedy) under identical runtime settings; MAT follows Sec. 3. CUDA events are synchronized at measurement points to ensure consistent timing.

## B  FULL ABLATION RESULTS PER TASK

Table 4 expands Table 2 by reporting task-wise MAT and Speedup under different combinations of causal-consistency path search (CPS) and adaptive draft-length (ADL).

| CPS | ADL | MT | | Trans | | Sum | | QA | | Math | | RAG | |
|-----|-----|------|------|------|------|------|------|------|------|------|------|------|------|
| | | MAT | Spd | MAT | Spd | MAT | Spd | MAT | Spd | MAT | Spd | MAT | Spd |
| ✓ | ✓ | **7.02** | **3.12×** | **7.35** | **3.40×** | **6.25** | 2.45× | **7.49** | **3.05×** | **7.61** | **4.05×** | **8.04** | 2.40× |
| ✓ | ✗ | 6.98 | 3.01× | 7.29 | 3.28× | 6.18 | 2.37× | 7.40 | 2.92× | 7.55 | 3.88× | 7.96 | 2.34× |
| ✗ | ✓ | 6.41 | 2.70× | 6.72 | 2.79× | 5.88 | 2.11× | 6.92 | 2.55× | 7.00 | 3.11× | 7.25 | 2.14× |
| ✗ | ✗ | 6.03 | 2.65× | 6.48 | 2.61× | 5.72 | 1.96× | 6.75 | 2.41× | 6.82 | 2.95× | 7.08 | 2.02× |

Table 4: **Task-wise ablation of DiffuSpec components.** CPS = causal-consistency path search; ADL = adaptive draft-length. Both components improve MAT and speedup (Spd) across tasks; **Spd denotes Speedup** ($\times$ vs. AR, $\uparrow$).

The task-wise breakdown confirms the complementary roles of CPS and ADL. CPS consistently yields larger gains, especially on QA and Math where alignment with AR verification is critical. ADL offers steady improvements by preventing over/under-drafting, with a visible effect on Summarization. Combining both mechanisms produces the best overall results, robust across all tasks.

## C  FIXED DRAFT LENGTH STUDY

For the fixed-$k$ study, we set $k_t \equiv k \in \{10, 20, 30, 50, 100\}$ at every decoding step and run CPS on the diffusion token lattice truncated to the first $k$ positions; the AR verifier then performs standard block verification on the resulting path, so the accepted prefix at each step has length $\leq k$.

We evaluate fixed proposal lengths $k \in \{10, 20, 30, 50, 100\}$ as well as the adaptive controller (ADL). Table 5 shows the trade-off: longer drafts increase acceptance length but reduce throughput due to higher rejection rates and drafting overhead.

| Policy | $k{=}10$ | $k{=}20$ | $k{=}30$ | $k{=}50$ | $k{=}100$ | **ADL** |
|---|---|---|---|---|---|---|
| Mean-MAT | 5.53 | 6.56 | 6.49 | 6.51 | 6.69 | **6.99** |
| Mean-Speedup | 2.74× | 2.98× | 2.98× | 2.91× | 2.78× | **3.08×** |

Table 5: **Fixed-$k$ vs. adaptive proposal length (quality-locked).** Means are computed across all tasks. ADL achieves the best speedup while also reaching the highest MAT, indicating a better speed–acceptance trade-off than fixed-$k$ policies.

As $k$ increases from 10 to 100, Mean-MAT generally rises (peaking at $k{=}100$ with 6.69), but Mean-Speedup peaks earlier at $k{=}20/30$ (both 2.98×) and then declines due to higher drafting and rejection costs. The adaptive controller (ADL) balances this trade-off online, attaining both the highest Mean-MAT (6.99) and the strongest Mean-Speedup (3.08×). This confirms the benefit of dynamic proposal sizing over fixed-$k$ policies.

## D  VOCABULARY MISMATCH AND HETEROGENEOUS TOKENIZERS

In our main experiments, the drafter and target share the same tokenizer. In practice, speculative decoding often needs to operate under *heterogeneous* vocabularies, where the drafter and target use different tokenizers. A naïve implementation can then become inefficient or even fail, since drafted tokens must be mapped into the target's vocabulary. This challenge has been studied in the SPS setting, and heterogeneous-vocabulary speculative decoding is now integrated into standard implementations in `transformers` (Timor et al., 2025).

DiffuSpec, like SPS, interacts with the target model only through the drafter's output probabilities and the target's next-token probabilities; it does not require access to hidden states or KV caches. As a result, DiffuSpec can directly reuse the same heterogeneous-vocabulary adapter at the probability interface without changing its core algorithm.

To illustrate this, we evaluate a setting where the drafter and target differ in both architecture and vocabulary: the target is Qwen3-32B, while the drafter is either Qwen2.5-7B (for SPS) or Dream-7B (for DiffuSpec). Both methods use the same heterogeneous-vocabulary adapter to bridge the drafter and target tokenizers. Table 6 reports results on Spec-Bench under this heterogeneous-vocabulary configuration.

| Method / Target = Qwen3-32B | Speedup (× vs. AR, ↑) | | | | | | Mean (MAT / Speedup) |
|---|---|---|---|---|---|---|---|
| | MT | Trans | Sum | QA | Math | RAG | |
| SPS (Qwen2.5-7B drafter) | 1.05× | 1.11× | 0.89× | 0.96× | 1.16× | 0.99× | 2.91 / 1.03× |
| DiffuSpec (Dream-7B drafter) | 1.23× | 1.66× | 0.96× | 1.40× | 2.17× | 1.05× | 3.02 / 1.42× |

Table 6: **Heterogeneous-vocabulary setting.** Target is Qwen3-32B; SPS uses Qwen2.5-7B as the drafter, and DiffuSpec uses Dream-7B. Both methods employ the same heterogeneous-vocabulary adapter at the probability interface. Speedup is measured relative to Qwen3-32B greedy decoding with its own tokenizer.

Even under heterogeneous vocabularies, DiffuSpec consistently yields higher speedups than SPS (Mean-Speedup 1.42× vs. 1.03×) while maintaining comparable Mean-MAT (3.02 vs. 2.91). This indicates that diffusion-based drafting remains effective when the drafter and target do not share a vocabulary, and that future advances in heterogeneous-vocabulary SPS can be directly adopted by DiffuSpec, since it only relies on the probability interface between drafter and target.

# E  OUTPUT VISUALIZATIONS

We provide qualitative $k$-sweeps showing how draft length shapes proposal style: short drafts tend to be terse; moderate drafts begin to exhibit step-by-step reasoning; very long drafts may drift or repeat. (All runs use the same prompt; only $k$ varies. Visualization samples are raw drafts before CPS/verification and are not correctness-checked.)

> **Algebra:** *Solve for x: $3x + 5 = 20$*
>
> $k$=6: Subtract 5 and divide by 3.
> $k$=60: To solve for $x$, subtract 5 from both sides and then divide by 3 to isolate $x$.
> $k$=200: To solve for $x$, we first isolate the variable by removing constants and normalizing the coefficient of $x$ ...

> **Word problem:** *A train travels at 60 km/h for 2.5 hours. How far?*
>
> $k$=6: 150 km.
> $k$=20: Distance = speed $\times$ time = $60 \times 2.5 = 150$ km.
> $k$=60: To find the distance, multiply the constant speed by the elapsed time; units remain in km.

> **Calculus:** *What is the derivative of $f(x) = 3x^2 + 2x + 1$?*
>
> $k$=6: $6x + 1$
> $k$=30: The derivative of $f(x) = 3x^2 + 2x + 1$ is $6x + 2$.
> $k$=40: The derivative of f(x) = 3x^2 + 2xx + 1 is ' ( xxx

As $k$ increases, drafts shift from terse answers to step-by-step reasoning (often with emerging chain-of-thought), which *initially* raises the verifier's accepted length: MAT grows for small-to-moderate $k$. Beyond a task-dependent sweet spot, however, we observe a clear *plateau*: very long drafts tend not to yield longer accepted prefixes—diffusion proposals begin to drift, repeat, or include partial phrases, so the AR verifier rejects earlier. Consequently, end-to-end speedup drops due to extra drafting and residual resampling, even though the draft itself is longer. This motivates *adaptive proposal sizing* (ADL) to stay near the knee of the MAT/speed trade-off, and *causal-consistency path search* (CPS) to keep proposals informative yet easy for the verifier to accept.

# F  THE USE OF LARGE LANGUAGE MODELS (LLMS)

We used off-the-shelf LLMs solely for language polishing (grammar and wording). No datasets, results, or analyses in this paper were generated by LLMs. All technical content, experiments, and conclusions were produced and verified by the authors.

