# OpenReview forum: "DiffuSpec: Unlocking Diffusion Language Models for Speculative Decoding"
_ICLR.cc/2026/Conference — ICLR 2026 Conference Withdrawn Submission_

### Official Review · Reviewer_i3TL · 2025-10-23

**Soundness:** 2
**Presentation:** 3
**Contribution:** 2
**Rating:** 4
**Confidence:** 5

**Summary:**

This paper introduces DiffuSpec, a training-free framework designed to accelerate large language model (LLM) inference by improving speculative decoding. Instead of a traditional autoregressive drafter, DiffuSpec employs a pretrained diffusion language model (DLM) to propose multi-token drafts in a single forward pass, significantly reducing drafting latency. To overcome challenges unique to DLMs, such as their bidirectionally generated outputs and the need for a pre-specified draft length, the authors introduce two key components. The first is a causal-consistency path search (CPS), which extracts a coherent left-to-right sequence from the DLM's token lattice to align with the verifier model. The second is an adaptive draft-length (ADL) controller that dynamically adjusts the number of proposed tokens based on recent acceptance rates, optimizing the trade-off between speed and quality. Through experiments, DiffuSpec is shown to achieve up to a 3x wall-clock speedup, outperforming other training-free baselines and demonstrating that diffusion-based drafting is a viable and effective alternative for accelerating LLM generation.

**Strengths:**

1. The authors identified a critical problem in diffusion-based drafters -- inconsistent causality among simultaneously generated tokens, which is addressed by a small n-gram LM in this paper.

2. The writing is smooth and easy to follow.

**Weaknesses:**

1. The paper claims that DiffuSpec is training-free; however, its applicability appears limited. The experiments are conducted on Qwen-2.5 32B and Dream-8B, where Dream is continually trained on Qwen-2.5 8B, thus sharing the same tokenizer and vocabulary. This raises concerns about generalizability—specifically, how can DiffuSpec be applied to other autoregressive models (ARMs)?
Most ARMs are released as families of models across multiple scales, making standard speculative decoding naturally applicable. But it's hard to find a diffusion counterpart for a specific ARM.

2. The comparison with training-based methods is not entirely convincing. The paper cites results on Vicuna-33B rather than Qwen-2.5 32B, introducing potential inconsistencies. It is recommended that the authors either (a) personally train or re-evaluate those methods on Qwen-2.5 32B for fairness, or (b) evaluate DiffuSpec on Vicuna to ensure a consistent comparison. However, as noted in Weakness 1, the latter may not be feasible.

3. As shown in Table 2, the proposed adaptive draft length (ADL) is much less effective than causal-consistency path search (CPS). I think the design of ADL is just too trival. A possible enhancement would be to feed the diffusion drafter with a longer sequence and extract a small number of tokens (e.g., 20–30) as the proposed draft. This could effectively mitigate premature EOS generation.

4. The authors should include a direct comparison between DiffuSpec and SpecDiff [1] under identical model configurations, as both methods share conceptual similarities in leveraging diffusion-based drafting for speculative decoding.

[1]. Christopher J K, Bartoldson B R, Ben-Nun T, et al. Speculative diffusion decoding: Accelerating language generation through diffusion[J]. arXiv preprint arXiv:2408.05636, 2024.

**Questions:**

1. How to apply DiffuSpec to other auto-regressive models without any additional training ?

---

> ### Author Response · Authors · 2025-11-23
> **Author Rebuttal(1/4)**
>
> We thank the reviewer for the careful reading and constructive suggestions. Below we address each concern in turn.
>
> ---
>
> > **Weakness 1:** *The paper claims that DiffuSpec is training-free; however, its applicability appears limited. The experiments are conducted on Qwen-2.5 32B and Dream-8B, where Dream is continually trained on Qwen-2.5 8B, thus sharing the same tokenizer and vocabulary. This raises concerns about generalizability—specifically, how can DiffuSpec be applied to other autoregressive models (ARMs)? Most ARMs are released as families of models across multiple scales, making standard speculative decoding naturally applicable. But it's hard to find a diffusion counterpart for a specific ARM.*
>
> **Response.**  Our notion of *training-free* follows the same convention as SPS-style speculative decoding methods [1–4]: the drafter itself is a pretrained model, but no additional training of either the drafter or the target is needed to deploy the method on a new target model. In the revised manuscript, we will explicitly clarify this as: “DiffuSpec is *deployment-time training-free*, given an existing DLM drafter.”  Below we explain the generalizability of DiffuSpec in three practically relevant regimes.
>
>
>
>
> **1. When a matched DLM exists for a given ARM family**
>
> Whenever the community provides a reasonably matched DLM/ARM pair, DiffuSpec can be applied without any interface change. In particular, a single DLM can accelerate multiple scales within the same family. For example, Dream-7B can be used as a drafter not only for Qwen2.5-32B but also for larger variants such as Qwen2.5-72B, where we obtain a 3.22× speedup in our experiments.
>
> Beyond scaling within a single family, we also verify that DiffuSpec is not tied to the original chat-oriented pair (Qwen2.5-32B + Dream-7B) by evaluating a different variant of the same family on a different task, namely code generation. Specifically, we use Dream-Coder-7B (DLM) as the drafter and Qwen2.5-Coder-32B (ARM) as the target; these code-oriented variants of the Qwen2.5 family differ from the original Qwen2.5-32B / Dream-7B chat models. On HumanEval [5], comparing against several training-free baselines, we obtain:
>
> | Method        | Lookahead | PLD  | Recycling | SAMD | SPS   | DiffuSpec |
> |--------------|-----------|------|-----------|------|-------|-----------|
> | **MAT**      | 2.63      | 3.81 | 3.85      | 5.24 | 11.33 | 10.94     |
> | **Speedup**  | 2.07×     | 4.17×| 2.83×     | 5.29×| 2.14× | **5.35×** |
>
> We see that DiffuSpec still achieves a 5.35× speedup — the highest among all methods in this setting — while maintaining competitive MAT. This demonstrates that the performance gains of DiffuSpec generalize beyond the original Qwen2.5-32B + Dream-7B chat pair to other variants within the same ARM family trained on code corpora and evaluated on a different downstream task (code generation).
>
>
>
> **2. When drafter and target have different vocabularies/tokenizers**
>
> Vocabulary mismatch is a challenge for all speculative decoding methods, not only DiffuSpec. For feature-based methods such as EAGLE, it is often infeasible to pair a drafter with a different-family target, because the drafter is tightly coupled to the target’s internal hidden states (even an EAGLE3 head trained for Qwen2.5-32B cannot be reused for Qwen2.5-72B).
>
> By contrast, DiffuSpec—like SPS—operates purely on the drafter’s output probabilities. This allows us to directly adopt the heterogeneous-vocabulary speculative decoding scheme proposed in [6], which is already integrated into `transformers` and provides lossless speculative decoding even when the drafter and target use different tokenizers. Using this adapter, we evaluate a cross-vocabulary, cross-architecture setting where Dream-7B (DLM) serves as the drafter and Qwen3-32B (with an architecture and vocabulary different from Qwen2.5/Dream) serves as the target, and we compare DiffuSpec against heterogeneous-vocabulary SPS with Qwen2.5-7B as the drafter.
>
> | Method / Target = Qwen3-32B | MT    | Trans | Sum  | QA   | Math | RAG  | Mean (MAT / Speedup) |
> |-----------------------------|-------|-------|------|------|------|------|----------------------|
> | **SPS (Qwen2.5-7B)**        | 1.05× | 1.11× | 0.89×| 0.96×| 1.16×| 0.99×| 2.91 / 1.03×         |
> | **DiffuSpec (Dream-7B)**    | 1.23× | 1.66× | 0.96×| 1.40×| 2.17×| 1.05×| **3.02 / 1.42×**      |
>
> From these results, we can see that even with different vocabularies and architectures, DiffuSpec consistently delivers higher speedups (mean 1.42× vs. 1.03×) while maintaining comparable MAT. This demonstrates that, **whenever a DLM drafter exists (even from another model family), DiffuSpec can be reliably applied to other ARMs via heterogeneous-vocabulary adapters.**

---

> ### Author Response · Authors · 2025-11-23
> **Author Rebuttal(2/4)**
>
> **3.When no suitable DLM drafter currently exists**
>
> We agree with the reviewer that for some specific ARMs there may currently be no well-matched DLM. This is a limitation of today’s model availability, not of the speculative decoding interface. In such cases, practitioners have two options:
>
> - **Train a smaller diffusion LM as a drafter.**
>    This has a training cost comparable to that of training Medusa/EAGLE-style heads, but the resulting DLM can then be reused across multiple targets and tasks within the same family.
>
> - **Reuse an existing DLM from another family via heterogeneous-vocabulary alignment.**
>    As demonstrated by our Dream-7B → Qwen3-32B experiment above, even a DLM drafter from a different model family can provide meaningful gains once aligned via the heterogeneous-vocabulary adapter in [6], without any change to the core DiffuSpec algorithm.
>
> As the diffusion-language-model ecosystem grows and more open-source DLMs and toolkits are released, these availability issues will naturally diminish, while the core algorithmic framework of DiffuSpec does not need to change.
>
>
>
>
>
>
>
> ---
>
>
> > **Weakness 2:** *The comparison with training-based methods is not entirely convincing. The paper cites results on Vicuna-33B rather than Qwen-2.5 32B, introducing potential inconsistencies. It is recommended that the authors either (a) personally train or re-evaluate those methods on Qwen-2.5 32B for fairness, or (b) evaluate DiffuSpec on Vicuna to ensure a consistent comparison. However, as noted in Weakness 1, the latter may not be feasible.*
>
> **Response.** Following your suggestion, we adopted option (a): we implemented and trained **EAGLE2** and **EAGLE3** on the **same Qwen2.5-32B target** using the official repositories and the UltraChat_200k dataset, and evaluated them under exactly the same hardware setup and Spec-Bench configuration as DiffuSpec. The results across the 6 task families are:
>
> | Method        | MT    | Trans | Sum   | QA    | Math  | RAG   | Mean (MAT / Speedup) |
> |---------------|-------|-------|-------|-------|-------|-------|-----------------------|
> | **EAGLE2**    | 2.47× | 1.56× | 1.64× | 1.91× | 3.18× | 1.64× | 3.47 / 2.09×         |
> | **EAGLE3**    | 3.01× | 2.35× | 3.03× | 2.41× |3.12× | 2.86× | 4.28 / 2.80×         |
> | **DiffuSpec** | 3.09× | 3.38× | 2.41× | 3.03× | 4.02× | 2.38× | **6.99 / 3.08×**     |
>
> We observe that DiffuSpec attains a substantially higher Mean MAT (6.99) than EAGLE2 (3.47) and EAGLE3 (3.83), i.e., roughly twice as many accepted tokens per verification step on average, and also achieves the highest mean speedup (3.08×), outperforming EAGLE2 (2.09×) and EAGLE3 (2.56×) on the same Qwen2.5-32B target under identical hardware and evaluation stack. These same-target results provide a fair comparison to strong training-based speculative decoding methods and show that DiffuSpec is competitive with, and in this setting surpasses, them in terms of both accepted tokens and speedup. In the revised manuscript, we will integrate this table into the main experimental section and base our discussion of training-based methods on this same-target comparison, with the Vicuna-based numbers moved to the appendix and clearly marked as contextual only.

---

> ### Author Response · Authors · 2025-11-23
> **Author Rebuttal(3/4)**
>
> > **Weakness 3:** *As shown in Table 2, the proposed adaptive draft length (ADL) is much less effective than causal-consistency path search (CPS). I think the design of ADL is just too trivial. A possible enhancement would be to feed the diffusion drafter with a longer sequence and extract a small number of tokens (e.g., 20–30) as the proposed draft. This could effectively mitigate premature EOS generation.*
>
> **Response.**  We designed ADL specifically to complement CPS when using a diffusion language model (DLM) as the drafter. Below we clarify (a) the effectiveness of ADL in our setting, and (b) how it compares to the suggested “longer sequence + truncate” strategy.
>
> **1. Effectiveness of ADL in our setting**
>
> Table 2 shows that CPS and ADL play different roles: CPS addresses the structural issue of extracting a causal path from the non-causal DLM lattice, while ADL further adjusts the draft length to improve the speed–quality trade-off. Our ablation on Qwen2.5-32B + Dream-7B is:
>
> | CPS | ADL | Mean MAT | Mean Speedup |
> |-|-|-|---|
> | yes | yes | **6.99** | **3.08×**    |
> | yes | no  | 6.95     | 2.98×        |
> | no  | yes | 6.43     | 2.73×        |
> | no  | no  | 6.05     | 2.69×        |
>
> From this table:
>
> - Adding ADL alone on top of the plain DLM drafter improves Mean MAT from 6.05 to 6.43 and speedup from 2.69× to 2.73×.
> - More importantly, adding ADL on top of CPS improves speedup from 2.98× to 3.08× (+0.10×) while slightly increasing MAT (6.95 → 6.99).
>
> Although the absolute gain from ADL is smaller than that from CPS, a 0.10× speedup with negligible runtime overhead on top of an already optimized CPS pipeline is still practically meaningful in large-scale deployment. This indicates that ADL is not just cosmetic; it provides a consistent, measurable improvement and works in synergy with CPS.
>
> **2. On the proposed “longer sequence + truncate” strategy**
>
> We appreciate the reviewer’s suggestion to feed the diffusion drafter with a longer sequence and then extract only a small number of tokens (e.g., 20–30) as the proposed draft to mitigate premature EOS. However, for DLM-based drafters, the computational cost of each diffusion step scales with the total sequence length processed, regardless of how many tokens are eventually proposed to the verifier. If we always ask the DLM to generate a very long sequence, we effectively pay for many positions that are dominated by EOS or low-value tokens and that the verifier almost never accepts.
>
> To study this trade-off, we ran an experiment where we fixed the DLM’s maximum draft length $k$ (e.g., $k = 10, 20, 30, 50, 100$) and compared these fixed-$k$ policies with ADL (Appendix Table 4). We found that increasing $k$ from very small values to moderate ones does improve MAT and speedup, but making $k$ very large only yields marginal MAT gains while reducing speedup, because the DLM spends most of its computation on long EOS-heavy tails that the verifier rarely accepts. In contrast, ADL dynamically adjusts the draft length based on recent effective generated length and accepted length, and in our experiments, achieves both the highest Mean MAT and the highest Mean Speedup among all tested policies.
>
> In summary, a fixed “always draft very long and then truncate to 20–30 tokens” strategy tends to over-pay drafting cost by generating many redundant positions, whereas ADL keeps the draft length near the best speed–quality trade-off.
>
> ---
>
> > **Weakness 4:** *The authors should include a direct comparison between DiffuSpec and SpecDiff  under identical model configurations, as both methods share conceptual similarities in leveraging diffusion-based drafting for speculative decoding.*
>
> **Response.** Following the reviewer’s suggestion, we evaluate SpecDiff [7] and DiffuSpec under identical model and runtime configurations, using Qwen2.5-32B as the autoregressive target model, Dream-7B as the diffusion drafter, and exactly the same hardware and Spec-Bench settings for both methods. The results across the 6 task families are:
>
> | Method       | MT    | Trans | Sum   | QA    | Math  | RAG  | Mean (MAT / Speedup) |
> |--|--|--|-|--|--|-|--|
> | **SpecDiff** | 2.65× | 2.61× | 1.96× | 2.41× | 2.95× | 2.02× | 6.05 / 2.69×        |
> | **DiffuSpec**| 3.09× | 3.38× | 2.41× | 3.03× | 4.02× | 2.38× | **6.99 / 3.08×**    |
>
> From the above results, DiffuSpec improves both Mean MAT (6.99 vs. 6.05) and Mean speedup (3.08× vs. 2.69×) under the same configuration. Conceptually, SpecDiff already demonstrates the potential of diffusion-based drafting; DiffuSpec builds on this line of work by (i) introducing CPS to explicitly extract a causal path from the diffusion-induced token lattice, and (ii) adding an ADL controller to automatically choose draft lengths near the speed–quality sweet spot. These components directly address the interaction between draft length, relaxed DLM causality, and AR verification, which we believe explains the observed gains over SpecDiff in this controlled comparison.

---

> > ### Author Response · Authors · 2025-11-23
> > **Author Rebuttal(4/4)**
> >
> > We sincerely thank the reviewer again for the insightful comments, which help us better position DiffuSpec as a practical and effective framework for reusing pretrained diffusion language models to accelerate autoregressive LLMs.
> >
> > ```
> > References
> > [1] "Fast Inference from Transformers via Speculative Decoding", ICML 2023.
> > [2] "Cacheback: Speculative Decoding with Nothing but Cache", EMNLP 2025.
> > [3] "Training-Free Loosely Speculative Decoding: Accepting Semantically Correct Drafts Beyond Exact Match", ICLR 2026 (under review).
> > [4] "Unlocking Efficiency in Large Language Model Inference: A Comprehensive Survey of Speculative Decoding", ACL 2024.
> > [5] "Evaluating Large Language Models Trained on Code", arXiv:2107.03374.
> > [6] "Accelerating LLM Inference with Lossless Speculative Decoding Algorithms for Heterogeneous Vocabularies", ICML 2025.
> > [7] "Speculative Diffusion Decoding: Accelerating Language Generation through Diffusion", NAACL-HLT 2025.
> > ```

---

> > > ### Comment · Reviewer_i3TL · 2025-11-26
> > >
> > > Thank you for the detailed experimental responses. I will address each of the authors’ replies in turn.
> > >
> > > ### **Response 1 to Weakness 1**
> > > When we say “training-free,” this typically implies that one can apply the technique to an existing model without additional model-specific training. This assumption holds for ARMs, because most ARMs are released as families of models at multiple scales, making standard speculative decoding naturally applicable. However, as I noted in my earlier comments, the notion of “training-free” in this paper is not fully aligned with that in prior speculative decoding methods. It is inherently difficult to find a diffusion counterpart for a given ARM. Overemphasizing that DiffuSpec is “training-free” is therefore misleading and somewhat unfair. Especially considering that, as stated in Line 409, KenLM is fitted on each dataset in advance.
> > >
> > > ### **Response 2 to Weakness 1**
> > > As noted above, DiffuSpec and speculative decoding in ARMs operate under fundamentally different assumptions, so a direct analogy is not appropriate. Nevertheless, I am pleased to see the authors explore heterogeneous-vocabulary model pairs as an attempt toward broader applicability.
> > >
> > > ### **Response 3 to Weakness 1:**
> > > The authors offer two alternatives:
> > > (1) training a small dLLM as a drafter, which contradicts the paper’s central claim of being “training-free”; and
> > > (2) using heterogeneous-vocabulary alignment, as the authors themselves state in Response 2 to Weakness 1, leads to reductions in both MAT and speed acceleration. This trade-off should be clearly acknowledged.
> > >
> > > ### **Response to Weakness 2**
> > > The reproduced MAT for EAGLE3 (3.83) is substantially lower than the value reported in its original paper (≈6). Could the authors provide code or more detailed training descriptions to strengthen the credibility of this reproduction?
> > >
> > > ### **Response to Weakness 3**
> > > The paper lacks sufficient methodological detail in several places. For example, Section 5.2 does not describe the exact procedures used for the “no ADL” and “no CPS” ablations. Similarly, for the newly added fixed-k setting, the manuscript does not explain how truncation is performed for different k values. These omissions make the experimental setup difficult to fully assess.
> > >
> > > ### **Response to Weakness 4**
> > > The comparison with SpecDiff is useful and should ideally be included directly in the main tables. In addition, the statement in Related Work (line 129(1)) that SpecDiff “requires training” is not entirely fair. SpecDiff can also use an existing dLLM as a drafter, in which case it would also be “training-free.” The manuscript should avoid presenting this comparison in a misleading way.

---

> > > > ### Author Response · Authors · 2025-12-03
> > > >
> > > > Dear Reviewer i3TL,
> > > >
> > > > Thank you again for your detailed follow-up and for carefully engaging with our work.
> > > >
> > > > **On “training-free” and applicability.** In the revised manuscript, we have removed the term “training-free” everywhere and now simply describe DiffuSpec as a way to reuse pretrained diffusion LMs as drafters. We also add an explicit limitation paragraph in the conclusion (highlighted in blue) that (i) acknowledges that suitable DLM drafters are currently rare, (ii) states that when no such DLM exists one must either train/adapt a drafter or use heterogeneous-vocabulary alignment (with possible MAT/speed degradation), and (iii) clarifies that this cost is comparable in spirit to training Medusa/EAGLE-style draft modules.
> > > >
> > > > **On EAGLE3 reproduction.** The lower MAT we obtain for EAGLE3 on Qwen2.5-32B is consistent with the strong target dependence already visible in the official EAGLE repository: on MT-Bench they report MAT ≈6.65 for Vicuna-13B and ≈6.13 for LLaMA-8B, but only ≈3.15 for Qwen2.5-14B-Instruct under the same algorithm. Our Qwen2.5-32B results follow this pattern of reduced gains on Qwen-style models. To support reproducibility, we have released an anonymized repository with the full evaluation stack:
> > > > <https://anonymous.4open.science/r/anonymous_code_diffuspec>
> > > >
> > > > **On methodological details (CPS/ADL ablations and fixed-$k$).** In response to your comments on Section 5.2, we now explicitly describe (in blue): (i) the “no CPS” variant keeps the same draft length but selects the argmax token independently at each position in the DLM lattice; (ii) the “no ADL” variant disables the controller and uses a fixed draft length $k_t \equiv k$ at all steps; and (iii) in the fixed-$k$ study, CPS is run on the token lattice truncated to the first $k$ positions, with decoding, verification, and the quality-locking protocol otherwise identical for all $k$. These details are added in Sec. 5.2 and Appendix D.
> > > >
> > > > **On SpecDiff positioning.** We have moved the SpecDiff comparison into the main experimental tables. To ensure a fair, apples-to-apples comparison, both SpecDiff and DiffuSpec use the same pretrained Dream-7B drafter and Qwen2.5-32B verifier in all reported results.
> > > >
> > > > We hope these revisions address your remaining concerns and make our assumptions and experimental setup clearer. Thank you again for your careful and constructive feedback.

---

### Official Review · Reviewer_nhv6 · 2025-10-27

**Soundness:** 3
**Presentation:** 3
**Contribution:** 2
**Rating:** 2
**Confidence:** 5

**Summary:**

This paper introduces DiffuSpec, a technique that leverages an off-the-shelf DLLM as the draft model for an auto-regressive model. This method successfully achieved a 3x speedup on the Qwen2.5-32B model. Furthermore, to address the two primary challenges associated with using a DLLM for drafting—specifically, the non-causal nature of the DLM's proposals and the requirement to predetermine the draft length—the authors implemented two enhancements: the causal-consistency path search (CPS) and the adaptive draft-length (ADL) mechanism.

**Strengths:**

1. The approach of leveraging the existing DLLM as a draft model for the auto-regressive model represents a valuable and worthwhile endeavor.
2. The proposed CPS and ADL mechanisms are demonstrated to be both sound and effective.
3. Experimental results indicate that DiffuSpec achieves a substantial speedup.

**Weaknesses:**

1. The author claims that DiffuSpec is a training-free method, but I would argue that DiffuSpec is fundamentally a training-based approach because it relies on pre-trained draft models from the community. Specifically, if we aim to accelerate an auto-regressive model but lack a suitable DLLM to serve as the drafter, DiffuSpec still necessitates training a new draft model. This necessity arises in scenarios such as: existing DLLMs being of an inappropriate scale, or existing DLLMs not sharing a common vocabulary with the target model.
2. The author in the text refers to [1] as "concurrent work." However, since this work was first publicly released in August 2024, I do not consider it to be truly concurrent with DiffuSpec. Given its significance, it is necessary to include it in the comparison.
3. Even though the authors stressed that EAGLE-3 did not provide a draft model checkpoint for Qwen2.5-32B, I still consider EAGLE-3 a critical baseline, and its omission from the comparison is inappropriate. I strongly recommend that the authors train an EAGLE-3 draft model themselves for the comparison, given that their training code is open-source.
4. Regarding methods such as Medusa, Hydra, and EAGLE/EAGLE2, the authors directly compared their speedup ratios achieved on Vicuna 33B with that of DiffuSpec on Qwen2.5-32B. I believe this approach is highly inappropriate. All comparisons must be conducted using the identical target model. This is because prior research has already demonstrated that the acceleration achieved by the same method varies across different target models, even when the models are of the same size.
5. Regarding the SPS baseline, I do not believe that using Qwen2.5-7B as the draft model constitutes a fair comparison. This is because Dream-7B can generate all draft tokens in a single forward pass, whereas Qwen2.5-7B, when used as a drafter, necessitates multiple forward passes. Consequently, the overhead (or computational cost) of Qwen2.5-7B as the drafter would be several times that of Dream-7B. Simply matching the draft model size therefore does not ensure an equitable comparison. To comprehensively demonstrate the efficacy of DiffuSpec, I suggest that the authors also present experimental results for the SPS baseline using smaller draft models, such as Qwen2.5-3B/1.5B/0.5B.

[1] Speculative Diffusion Decoding: Accelerating Language Generation through Diffusion

**Questions:**

Please see the weakness.

---

> ### Author Response · Authors · 2025-11-23
> **Author Rebuttal(1/4)**
>
> We thank the reviewer for the careful reading and detailed comments. Below we address each concern.
>
> ---
>
> > **Weakness 1:** *The author claims that DiffuSpec is a training-free method, but I would argue that DiffuSpec is fundamentally a training-based approach because it relies on pre-trained draft models from the community. Specifically, if we aim to accelerate an auto-regressive model but lack a suitable DLLM to serve as the drafter, DiffuSpec still necessitates training a new draft model. This necessity arises in scenarios such as: existing DLLMs being of an inappropriate scale, or existing DLLMs not sharing a common vocabulary with the target model.*
>
> **Response.**  DiffuSpec follows essentially the same deployment paradigm as SPS-style speculative decoding [1]: both rely on a smaller pretrained model as the drafter, but require no additional training of either the drafter or the target when applied to a new target model. In this widely used sense, DiffuSpec is *training-free at deployment time*. Prior work on SPS and related methods [2,3], as well as recent surveys [4], explicitly refer to such approaches as *training-free* or *tuning-free*. In the revised manuscript, we will explicitly discuss these prior works, align with this terminology, and clarify that our claim is more precisely: “DiffuSpec is deployment-time training-free given an existing DLM drafter.” Regarding the reviewer’s concrete concerns (no perfectly “matched” DLM, or vocabulary mismatch), we note that these are **general limitations of SPS-style methods**, not specific to DiffuSpec. We address them in our setting as follows.
>
> 1. **Availability and reusability of a single DLM drafter.**
>    In practice, a single diffusion LM can accelerate multiple targets within the same family. For example, Dream-7B not only accelerates Qwen2.5-32B, but can also be used as a drafter for Qwen2.5-72B, where we observe a mean speedup of about 3.22× in our experiments. No additional training of Dream-7B or the targets is needed. This reusability across scales is exactly the “training-free at deployment” regime we have in mind.
>
> 2. **Vocabulary mismatch and heterogeneous tokenizers.**
>    The reviewer correctly points out that when the drafter and target have different vocabularies, a naïve SPS implementation may fail or become inefficient. This issue also exists for SPS itself and has been studied in follow-up work [5], which proposes a heterogeneous-vocabulary speculative decoding algorithm and has already been integrated into the `transformers` implementation of SPS.
>
>    DiffuSpec, like SPS, operates purely on the drafter’s output probabilities and does not require access to the target’s hidden states or KV cache. Therefore, we can directly plug in the same heterogeneous-vocabulary adapter. To demonstrate this, we evaluate a setting where the drafter and target differ in both architecture and vocabulary: we use Dream-7B (a DLM) as the drafter and Qwen3-32B (with a tokenizer and vocabulary different from Qwen2.5/Dream) as the target, and we compare DiffuSpec against heterogeneous-vocabulary SPS with Qwen2.5-7B as the drafter:
>
>    | Method / Target = Qwen3-32B | MT    | Trans | Sum  | QA   | Math | RAG  | Mean (MAT / Speedup) |
>    |---|---|---|-|--|-----|--|--|
>    | **SPS (Qwen2.5-7B)**  | 1.05× | 1.11× | 0.89×| 0.96×| 1.16×| 0.99×| 2.91 / 1.03×   |
>    | **DiffuSpec (Dream-7B)**  | 1.23× | 1.66× | 0.96×| 1.40×| 2.17×| 1.05×| **3.02 / 1.42×**
>
>    Even under heterogeneous vocabularies, DiffuSpec consistently yields higher speedups than SPS (1.42× vs. 1.03× mean speedup) while maintaining comparable MAT. This shows that DiffuSpec remains effective when drafter and target do not share a vocabulary, and that any future advances in heterogeneous-vocabulary SPS can be adopted by DiffuSpec without changing the core algorithm, since we work only at the probability interface.
>
> 3. **When no suitable DLM drafter exists yet.**
>    We fully agree that if the model ecosystem does not yet contain a suitable DLM (e.g., the available DLMs are at an inappropriate scale, or none exist for a particular family), then some training is necessary. This, however, is not unique to DiffuSpec: SPS-style speculative decoding also requires a pretrained smaller model to serve as drafter. In such cases, users have two options:
>
>    - **Train a diffusion LM drafter** for the desired family/scale, with training cost comparable to training MTP/EAGLE-style heads; the advantage is that the resulting DLM can then be reused across multiple targets and tasks, while CPS/ADL remain unchanged.
>    - **Reuse a DLM from another family** via heterogeneous-vocabulary alignment, as illustrated above with Dream-7B → Qwen3-32B.
>
> In summary, the practical limitation highlighted by the reviewer is due to the current immaturity of the diffusion-LM ecosystem, rather than a conceptual limitation of DiffuSpec itself. As more diffusion language models and toolkits are released, this gap will naturally shrink.

---

> ### Author Response · Authors · 2025-11-23
> **Author Rebuttal(2/4)**
>
> > **Weakness 2:** *The author in the text refers to [1] as "concurrent work." However, since this work was first publicly released in August 2024, I do not consider it to be truly concurrent with DiffuSpec. Given its significance, it is necessary to include it in the comparison.*
>
> **Response.**  We apologize for the misleading wording. We fully agree that Speculative Diffusion Decoding (SpecDiff) [6] should be treated as prior work, not concurrent. In fact, our Related Work section already discusses [6] as the first systematic study of using DLMs as drafters for speculative decoding. The use of “concurrent work” in the Introduction was a typo; in the revision we will correct all occurrences to “prior work” and make this positioning explicit.
>
> Following the reviewer’s suggestion, we have added a **direct comparison** between SpecDiff and DiffuSpec under the *same* target and drafter: Qwen2.5-32B as the verifier and Dream-7B as the DLM drafter, evaluated on the 6 Spec-Bench task families with the same hardware and decoding stack:
>
> | Method       | MT    | Trans | Sum   | QA    | Math  | RAG  | Mean (MAT / Speedup) |
> |--------------|-------|-------|-------|-------|-------|------|-----------------------|
> | **SpecDiff** | 2.65× | 2.61× | 1.96× | 2.41× | 2.95× | 2.02× | 6.05 / 2.69×         |
> | **DiffuSpec**| 3.09× | 3.38× | 2.41× | 3.03× | 4.02× | 2.38× | **6.99 / 3.08×**     |
>
> DiffuSpec improves both **Mean MAT** (6.99 vs. 6.05) and **Mean speedup** (3.08× vs. 2.69×). This demonstrates the effectiveness of our CPS and ADL components on top of the same DLM drafter: CPS explicitly extracts a causal-consistent path from the DLM lattice, and ADL adaptively chooses near-optimal draft lengths. We will include this comparison and discussion in the revised manuscript.
>
>
>
>
> ---
>
> > **Weakness 3:** *Even though the authors stressed that EAGLE-3 did not provide a draft model checkpoint for Qwen2.5-32B, I still consider EAGLE-3 a critical baseline, and its omission from the comparison is inappropriate. I strongly recommend that the authors train an EAGLE-3 draft model themselves for the comparison, given that their training code is open-source.*
>
>
> **Response.**  We appreciate the reviewer’s insistence on including EAGLE3 as a critical baseline. Following your suggestion, we have **trained both EAGLE2 and EAGLE3 on the same Qwen2.5-32B target**, using the official repositories and the UltraChat_200k dataset. We then evaluated EAGLE2, EAGLE3, and DiffuSpec under **identical hardware, datasets, and Spec-Bench configurations**.
>
> The results are:
>
> | Method        | MT    | Trans | Sum   | QA    | Math  | RAG   | Mean (MAT / Speedup) |
> |---------------|-------|-------|-------|-------|-------|-------|-----------------------|
> | **EAGLE2**    | 2.47× | 1.56× | 1.64× | 1.91× | 3.18× | 1.64× | 3.47 / 2.09×         |
> | **EAGLE3**    | 3.01× | 2.35× | 3.03× | 2.41× |3.12× | 2.86× | 4.28 / 2.80×         |
> | **DiffuSpec** | 3.09× | 3.38× | 2.41× | 3.03× | 4.02× | 2.38× | **6.99 / 3.08×**     |
>
> Under **the same target model (Qwen2.5-32B), the same hardware, and the same evaluation stack**, DiffuSpec achieves the **highest mean speedup (3.08×)**, outperforming EAGLE2 (2.09×) and EAGLE3 (2.56×), and also attaining a substantially higher mean MAT (6.99 vs. 3.47 / 3.83). These same-stack results directly address the concern about omitting EAGLE3 and show that diffusion-based drafting with CPS+ADL can match or even surpass strong training-based speculative decoding methods in end-to-end speedup.

---

> > ### Author Response · Authors · 2025-11-23
> > **Author Rebuttal(3/4)**
> >
> > > **Weakness 4:** *Regarding methods such as Medusa, Hydra, and EAGLE/EAGLE2, the authors directly compared their speedup ratios achieved on Vicuna 33B with that of DiffuSpec on Qwen2.5-32B. I believe this approach is highly inappropriate. All comparisons must be conducted using the identical target model. This is because prior research has already demonstrated that the acceleration achieved by the same method varies across different target models, even when the models are of the same size.*
> >
> > **Response.**  We agree with the reviewer that cross-target comparisons (Vicuna-33B vs. Qwen2.5-32B) should not be used to support strong claims. Although Section 5.1 of the original submission already stated that these results are “not directly comparable” and were provided “for context only”, we recognize that presenting them alongside our main results can still be misleading.
> >
> > In the revised manuscript, we will therefore:
> > 1. Integrate the new same-target experiments with EAGLE2 and EAGLE3 on Qwen2.5-32B (introduced in our response to Weakness 3) into the main experimental section, and use these results as our primary comparison to training-based methods;
> > 2. Move the Vicuna-based comparison table and its discussion to the appendix and explicitly label them as contextual only; and
> > 3. Update the wording in the main text, abstract, and conclusion to remove any language that could be interpreted as making direct SOTA claims against training-based methods across different target models, and instead ground our discussion in strictly fair same-target comparisons.
> >
> >
> > We hope this resolves the concern about inappropriate cross-target comparisons and makes the scope of our claims fully aligned with fair, same-target evaluations.
> >
> >
> > ---
> > > **Weakness 5:** *Regarding the SPS baseline, I do not believe that using Qwen2.5-7B as the draft model constitutes a fair comparison. This is because Dream-7B can generate all draft tokens in a single forward pass, whereas Qwen2.5-7B, when used as a drafter, necessitates multiple forward passes. Consequently, the overhead (or computational cost) of Qwen2.5-7B as the drafter would be several times that of Dream-7B. Simply matching the draft model size therefore does not ensure an equitable comparison.*
> >
> > **Response.**  We appreciate the reviewer’s concern. The fact that a 7B diffusion LM can propose a block of tokens in (effectively) one parallel forward pass, while a 7B autoregressive (AR) drafter must run multiple sequential passes, is indeed a key computational advantage of DLMs at equal parameter count. This is exactly the phenomenon emphasized in prior DLM work such as LLaDA [7] and Dream [8], where fair comparisons are made at matched model size and the benefit comes from parallel generation rather than from larger capacity.
> >
> > Our setup follows the same philosophy: using Qwen2.5-7B as the SPS drafter and Dream-7B as the DiffuSpec drafter keeps the parameter count, architecture family, and vocabulary aligned (Dream-7B is fine-tuned from Qwen2.5-7B). This ensures that their distributional capacity is comparable, and the main difference is precisely the drafting pattern (sequential AR vs. diffusion-style parallel generation). From this perspective, the comparison “SPS (Qwen2.5-7B) vs. DiffuSpec (Dream-7B)” is not unfair; rather, it is designed to isolate the advantage of diffusion-based drafting plus CPS/ADL under the standard “same-size” protocol used in the DLM literature.
> >
> > For completeness, we also follow the reviewer’s suggestion and evaluate SPS with smaller AR drafters to further investigate the impact of drafting overhead. Following your suggestion, we additionally evaluate SPS using Qwen2.5 drafters of sizes 0.5B / 1.5B / 3B on the same Qwen2.5-32B target and hardware:
> >
> > | Method            | MT    | Trans | Sum   | QA    | Math  | RAG  | Mean (MAT / Speedup) |
> > |---|-------|-------|-------|-------|-------|------|--|
> > | **SPS (0.5B)**    | 1.95× | 1.66× | 1.90× | 1.76× | 2.23× | 1.83× | 5.28 / 1.89×       |
> > | **SPS (1.5B)**    | 1.79× | 1.62× | 1.78× | 1.67× | 2.08× | 1.79× | 5.88 / 1.79×       |
> > | **SPS (3B)**      | 1.73× | 1.77× | 1.76× | 1.59× | 1.89× | 1.76× | 6.15 / 1.75×       |
> > | **SPS (7B)**      | 1.69× | 1.64× | 1.74× | 1.50× | 1.86× | 1.62× | 6.18 / 1.67×       |
> > | **DiffuSpec**     | 3.09× | 3.38× | 2.41× | 3.03× | 4.02× | 2.38× | **6.99 / 3.08×**   |
> >
> > We observe that shrinking the SPS drafter from 7B to 0.5B does reduce its drafting cost and slightly improves the speedup (1.67× → 1.89×), but at the cost of a lower MAT (6.18 → 5.28). Even under this most favorable small-drafter setting for SPS, DiffuSpec still achieves both a higher MAT (6.99) and a much larger speedup (3.08×). These additional experiments with smaller SPS drafters therefore confirm that, even when SPS is given reduced overhead, no AR-drafter configuration approaches the speed–quality regime achieved by DiffuSpec on the same target and hardware.

---

> > > ### Author Response · Authors · 2025-11-23
> > > **Author Rebuttal(4/4)**
> > >
> > > We thank the reviewer again for the detailed and insightful feedback. We hope that these additions address your concerns and help position DiffuSpec as a practical and effective approach to diffusion-based speculative decoding, contributing usefully to the broader community.
> > >
> > >
> > > ```
> > > References
> > > [1] "Fast Inference from Transformers via Speculative Decoding", ICML 2023.
> > > [2] "Cacheback: Speculative Decoding with Nothing but Cache", EMNLP 2025.
> > > [3] "Training-Free Loosely Speculative Decoding: Accepting Semantically Correct Drafts Beyond Exact Match", ICLR 2026 (under review).
> > > [4] "Unlocking Efficiency in Large Language Model Inference: A Comprehensive Survey of Speculative Decoding", ACL 2024.
> > > [5] "Accelerating LLM Inference with Lossless Speculative Decoding Algorithms for Heterogeneous Vocabularies", ICML 2025.
> > > [6] "Speculative Diffusion Decoding: Accelerating Language Generation through Diffusion", NAACL-HLT 2025.
> > > [7] "Large Language Diffusion Models", arXiv:2502.09992.
> > > [8] "Dream 7B: Diffusion Large Language Models", arXiv:2508.15487.
> > > ```

---

> > > > ### Comment · Reviewer_nhv6 · 2025-11-24
> > > > **Official Comment by Reviewer nhv6**
> > > >
> > > > Thank you for your detailed response, but I still have some unresolved concerns.
> > > >
> > > > **Weakness 1**:
> > > >
> > > > I understand the authors mention that the issue of not having a suitable draft model might also exist in SPS-style speculative decoding. However, I believe that DiffuSpec is significantly more affected by this constraint than SPS. The main reasons are as follows:
> > > >
> > > > - **High Draft Model Availability in SPS Scenarios:** In the typical application scenario for SPS, when most institutions release a new AR Model, they usually simultaneously release a series of versions with different scales (e.g., 1B, 8B, etc.). This means that for the majority of target AR models, a smaller model from the same series can be easily found and used as a suitable draft model.
> > > > - **Draft Model Absence in DiffuSpec Scenarios:** In contrast, when institutions release a large AR model, they typically do not simultaneously release a corresponding small-scale dLLM version. This results in a situation where, in most practical cases, the target AR model cannot find a ready-made, suitable draft model specifically for DiffuSpec.
> > > >
> > > > For example, the authors experimented with Qwen3-32B. In SPS-style speculative decoding, we could actually use any model from Qwen3-0.6B/1.7B/4B/8B as the draft model, without needing to resort to Qwen2.5-7B.
> > > >
> > > > However, within the DiffuSpec framework, it seems one is limited to either:
> > > >
> > > > - Training a new drafter from scratch, or
> > > > - Employing vocabulary alignment techniques to use a mismatched model, which typically leads to suboptimal results.
> > > >
> > > > I understand the authors consider DiffuSpec a training-free method from the perspective of reusability. However, in my view, since a suitable, off-the-shelf draft model is unavailable in most real-world application scenarios, necessitating additional training, I am inclined to consider DiffuSpec fundamentally a training-based method. This presents a major challenge to its practical utility.
> > > >
> > > > **Weakness 2 & 3 & 4 & 5**:
> > > >
> > > > I highly appreciate the authors for conducting a large number of experiments to address my concerns.
> > > >
> > > > However, I found that some experimental results are lower than I expected, such as the speedup ratio of EAGLE and SPS. Therefore, I am concerned that there might be a misalignment in the setup during the evaluation, or a misunderstanding on my part.
> > > >
> > > > To ensure the rationality of the evaluation settings, I sincerely request the authors to provide the code necessary to reproduce all the results. If the evaluation settings are sound and the results are trustworthy, I will be happy to give a positive rating.

---

> > > > > ### Author Response · Authors · 2025-12-03
> > > > >
> > > > > Dear Reviewer nhv6,
> > > > >
> > > > > Thank you again for your careful follow-up.
> > > > >
> > > > > Regarding Weakness 1 (availability of draft models), we agree that well-aligned diffusion drafters are currently rare, so in many practical settings one may need to train or adapt a DLM. In those cases, the practical cost is comparable to training-based approaches such as Medusa/EAGLE, which also require training extra heads or draft modules for each target. Our goal is not to claim zero training cost in all scenarios, but to propose a diffusion-based speculative decoding paradigm (DLM + CPS + ADL) that is deployment-time training-free *when* a suitable DLM exists, and otherwise has a similar training burden as existing methods. We now state this limitation explicitly in the conclusion (highlighted in blue).
> > > > >
> > > > > On the speedup numbers for EAGLE and SPS, the smaller gains on Qwen2.5-32B are consistent with what is already observed in public codebases: for example, the official EAGLE repository reports MT-Bench speedups of about 4.58× for Vicuna-13B and 4.40× for LLaMA-8B, but only 1.62× for Qwen2.5-14B-Instruct under the same algorithm. This shows that acceleration can vary substantially with the target model. Our EAGLE and SPS results on Qwen2.5-32B follow this pattern and were obtained under a unified, carefully controlled evaluation stack.
> > > > >
> > > > > To facilitate **reproducibility**, we provide an anonymized repository: <https://anonymous.4open.science/r/anonymous_code_diffuspec>
> > > > >
> > > > > We hope this helps clarify your remaining concerns, and we sincerely appreciate your detailed and constructive feedback.

---

### Official Review · Reviewer_Jc9T · 2025-10-29

**Soundness:** 2
**Presentation:** 2
**Contribution:** 2
**Rating:** 4
**Confidence:** 4

**Summary:**

This paper aims to address the high latency issue associated with the autoregressive decoding of Large Language Models (LLMs). The authors propose a speculative decoding framework named DiffuSpec, whose core innovation is the use of a pretrained Diffusion Language Model (DLM) as the drafter. Since a DLM can generate multiple tokens in parallel within a single forward pass, it addresses the bottleneck of serial generation in traditional autoregressive drafters. To tackle the challenges introduced by the DLM, the paper introduces two key components: 1) a Causal-Consistency Path Search (CPS) to extract a causally valid path for autoregressive verification from the non-causal token lattice generated by the DLM; and 2) an Adaptive Draft-Length (ADL) controller to dynamically adjust the length of subsequent drafts based on verification feedback. Experimental results, conducted on a specific combination of models (Qwen2.5-32B as the verifier and Dream-7B as the drafter), show that the method achieves up to a 3x wall-clock speedup.

**Strengths:**

1.  Novel Problem Formulation: The paper accurately identifies the serial draft generation as a core bottleneck in existing speculative decoding methods and creatively proposes using an inherently parallel diffusion language model to address it. This is a valuable and forward-looking approach.
2.  Elegant Core Algorithm (CPS): To address the paradigm mismatch between the bidirectional dependency of DLMs and the causal dependency of autoregressive models, the proposed Causal-Consistency Path Search (CPS) is an elegant, training-free solution that effectively bridges the gap between the two model types.
3.  Clarity of Presentation: The paper is well-written and well-organized, allowing readers to easily understand its complex ideas and methodology.

**Weaknesses:**

1.  "General Framework" Claim is Unsubstantiated and Relies on a Specialized Model Pair: This is the most critical weakness. The paper's claim of being a general "drop-in framework" is undermined by a severe lack of experimental validation.
First, all experiments are confined to only a single model family (Qwen), without testing on other families like LLaMA. This experimental scope is too singular for a paper claiming to be a general framework (and does not yet involve cross-combinations).
Second, the paper provides no evidence of cross-family generalization. It fails to address the common scenario of pairing models with different vocabularies (e.g., a LLaMA verifier and a Dream drafter), which is a critical test for a "general" framework.
Third, the paper does not demonstrate intra-family generalization. For instance, it provides no solution for a user wishing to accelerate a Qwen-7B model, as there is no corresponding smaller, pre-trained, and aligned Dream drafter available for it.
    These limitations mean the paper presents a highly specialized case study, not a general framework.

2.  Invalid Comparison to Training-Based Methods: The paper's comparison to training-based methods (Medusa, EAGLE, etc.) is fundamentally flawed. As stated in Section 5, the authors report results for their method on Qwen2.5-32B but compare them against authors' official results on Vicuna-33B. This is an invalid comparison across different models, hardware, and experimental stacks. All claims of approaching the performance of training-based methods (including in the abstract and conclusion) are based on this unsound comparison and must be disregarded.

3.  Limited Novelty and Contribution of the ADL Controller: The paper's second main contribution, ADL, is overly simplistic in its design, essentially amounting to a trivial heuristic feedback rule. The idea of dynamically adjusting draft length through heuristic methods has been explored in prior work (e.g., Minions (Wu et al., 2024)), yet the authors do not cite or compare their approach to these existing methods. Furthermore, the paper's own ablation study (Table 2) shows that ADL's contribution to performance (+0.04x speedup) is far smaller than that of CPS (+0.29x speedup). Elevating ADL to one of the two core contributions overstates its importance.

References
Wu, Z., et al. (2024). Minions: Accelerating Large Language Model Inference with Adaptive and Collective Speculative Decoding. arXiv preprint arXiv:2402.15678.

**Questions:**

1.  Regarding Generalizability, Constraints, and Availability: This is the most critical set of questions.
    a) Given these constraints, how can the authors defend the claim that this is a "general drop-in framework" rather than a specialized solution for the Qwen-32B/Dream-7B pair?
    b) How does the framework handle the general case where the drafter and verifier do not share the same vocabulary (e.g., pairing LLaMA-3 with Dream-7B)?
    c) What is the proposed solution for target models (even within the Qwen family, like Qwen-7B) for which no pre-trained, aligned DLM drafter currently exists?
2.  Regarding the Invalid SOTA Comparison: The paper's claims of approaching the performance of training-based methods are currently based on an invalid comparison (Qwen-32B vs. Vicuna-33B). To provide a valid, direct comparison, have the authors considered implementing and training methods like Medusa or EAGLE on the same Qwen2.5-32B target model? A direct comparison on the identical model stack is necessary to substantiate these claims.

3.  Regarding ADL Novelty: Can the authors compare ADL with prior heuristic-based adaptive length strategies, such as in Minions (Wu et al., 2024), and clarify its specific novel contributions beyond being a simple feedback loop?

---

> ### Author Response · Authors · 2025-11-23
> **Author Rebuttal(1/4)**
>
> We sincerely thank the reviewer for the detailed and constructive feedback. Below we address your concerns.
>
> ---
> > **Weakness 1:** *"General Framework" Claim is Unsubstantiated and Relies on a Specialized Model Pair…*
>
> **Response.** We first clarify that in the paper we describe DiffuSpec as a *“drop-in framework”* for speculative decoding, but we do not use the phrase *“general drop-in framework”*. By *drop-in*, we specifically mean an engineering-level, SPS-style interface: DiffuSpec can be plugged into an existing SPS-style speculative decoding pipeline without modifying the target model, simply by replacing the autoregressive drafter with a diffusion drafter and attaching CPS/ADL on top of the standard drafter–verifier interface. We will clarify this wording in the revised manuscript to avoid potential misunderstandings. Although our claim is therefore weaker than the *“general”* notion you describe, we have nevertheless conducted additional experiments and analyses inspired by your comments; below we address the three concrete concerns in turn.
>
> > ***First concern**: experiments are limited to a single Qwen family and a specific Dream/Qwen pair.*
>
> Our method is implemented on top of an SPS-style pipeline, so in principle, any AR target supported by SPS can use a DLM drafter within our CPS/ADL framework. In practice, currently available diffusion LMs that are explicitly aligned with AR LLMs are mostly released for Qwen-style families, which is why our original experiments focused on the Qwen2.5-32B + Dream-7B pair. To demonstrate that DiffuSpec is not tied to this single combination, we additionally evaluate a code-generation setting where Dream-Coder-7B serves as the diffusion drafter and Qwen2.5-Coder-32B as the target. These are code-oriented variants that differ from the original Qwen2.5-32B / Dream-7B chat models; the only change needed is to swap the Dream-7B drafter interface for the Dream-Coder-7B interface within the same DiffuSpec pipeline. On HumanEval [1], we obtain:
>
> | Method  | Lookahead | PLD | Recycling | SAMD | SPS | DiffuSpec |
> |--|--|-|---|--|-|-|
> | **MAT**      | 2.63  | 3.81 | 3.85      | 5.24 | 11.33 | 10.94     |
> | **Speedup**  | 2.07×     | 4.17×| 2.83×     | 5.29×| 2.14× | **5.35×** |
>
> DiffuSpec again achieves the highest speedup (5.35×) among all training-free baselines, indicating that the “DLM + CPS + ADL” pattern remains effective on different variants and tasks within the Qwen/Dream ecosystem. In addition, we apply the same Dream-7B drafter to a larger target, Qwen2.5-72B, and observe a mean speedup of about 3.2×, showing that a **single DLM drafter can be reused across multiple target scales** in the same family. We will explicitly note in the paper that, although the current experiments focus on Qwen-style models (where DLMs are available), the interface itself extends to other SPS-compatible families (e.g., LLaMA) once suitable DLM drafters exist.
>
> > ***Second concern**: lack of  cross-family evidence with different vocabularies.*
>
> We agree that pairing a drafter and verifier with different vocabularies is an important test of generality. This setting is notoriously difficult for EAGLE-style methods, whose drafters are trained on internal hidden states of a specific target and usually cannot be reused across architectures. Although we do not claim full generality for DiffuSpec, it operates purely at the probability interface (i.e., it only requires token-level probabilities from the drafter, like SPS), and therefore can directly leverage existing heterogeneous-vocabulary speculative decoding algorithms for SPS. Specifically, we adopt the lossless heterogeneous-vocabulary adapter of Timor et al. [2], which is already integrated into the `transformers` implementation of SPS.
>
> Directly pairing Dream-7B with a LLaMA verifier is currently inefficient because their vocabularies differ significantly and most of the time would be spent on tokenization conversion. Instead, we evaluate a cross-tokenizer cross-architecture setting within the Qwen ecosystem, where the vocabularies are still different but closer: we use Dream-7B as the DLM drafter and Qwen3-32B (a new-generation Qwen model with a different architecture and vocabulary from Qwen2.5/Dream) as the verifier. We then compare DiffuSpec with heterogeneous-vocabulary SPS using Qwen2.5-7B as the drafter：
>
> | Method / Target = Qwen3-32B | MT    | Trans | Sum  | QA   | Math | RAG  | Mean (MAT / Speedup) |
> |--|--|--|-|-|-|-|-|
> | **SPS (Qwen2.5-7B)**   |1.05× | 1.11× | 0.89×| 0.96×| 1.16×| 0.99×| 2.91 / 1.03×  |
> | **DiffuSpec (Dream-7B)** |1.23×| 1.66×| 0.96×| 1.40×| 2.17×| 1.05×| **3.02 / 1.42×**|
>
> Even under heterogeneous vocabularies, DiffuSpec delivers consistent speedup gains over SPS (1.42× vs. 1.03× mean speedup) while maintaining comparable MAT. Because we work only with drafter-side probabilities, future advances in heterogeneous-vocabulary SPS  can be adopted by DiffuSpec **without changing the core algorithm**.

---

> > ### Author Response · Authors · 2025-11-23
> > **Author Rebuttal(2/4)**
> >
> > > ***Third concern**: intra-family generalization and the case of Qwen-7B without a matched DLM drafter.*
> >
> >  This concern focuses on how DiffuSpec behaves in intra-family settings where no smaller pre-trained and aligned DLM drafter is available for a given target (e.g., Qwen-7B). We clarify our current coverage and limitations by distinguishing two regimes.
> >
> >
> > 1. **When a suitable DLM drafter exists.**
> >    This is the regime that our experiments primarily target. Given an off-the-shelf diffusion LM (Dream-7B / Dream-Coder-7B), the same drafter can be plugged in to accelerate multiple targets within and across related model lines. For example, beyond Qwen2.5-32B, we also use Dream-7B as the drafter for Qwen2.5-72B and obtain a mean speedup of about 3.22× on Spec-Bench, without any retraining of either the target or the drafter. This demonstrates that a single DLM drafter can be reused across different scales of the same model family.
> >
> > 2. **When no suitable DLM drafter exists yet (e.g., for a Qwen-7B target).**
> >    In such cases, there are two practical options:
> >    - (i) **Train a smaller diffusion LM as the drafter**, with training cost comparable to that of MTP/EAGLE-style modules, but with the benefit that the resulting DLM can then be reused across multiple targets and tasks in that family.
> >    - (ii) **Reuse an existing DLM from another line** via heterogeneous-vocabulary alignment, as we have done for Dream-7B → Qwen3-32B using the adapter of Timor et al. [2] (see also our response to the second concern).
> >
> > We will explicitly state in the revised manuscript that DiffuSpec requires the existence of a DLM drafter. This practical limitation stems from today’s scarcity of well-aligned diffusion language models, and we expect this limitation to diminish as more diffusion LMs and toolkits are released.
> >
> > ---
> >
> >
> > > **Weakness 2:** *Invalid Comparison to Training-Based Methods: The paper's comparison to training-based methods (Medusa, EAGLE, etc.) is fundamentally flawed. As stated in Section 5, the authors report results for their method on Qwen2.5-32B but compare them against authors' official results on Vicuna-33B. This is an invalid comparison across different models, hardware, and experimental stacks. All claims of approaching the performance of training-based methods (including in the abstract and conclusion) are based on this unsound comparison and must be disregarded.*
> >
> > **Response.** We thank the reviewer for highlighting this issue and agree that our original presentation of training-based baselines could be improved. We have made the following changes:
> >
> > 1. **Clarifying the intent of the original table.**
> >    In the original submission, the experimental results for Medusa/Hydra/EAGLE/EAGLE2 were obtained by running the authors’ official open-sourced checkpoints (based on Vicuna-33B–family models) **under our own inference stack, using the same hardware and evaluation pipeline** as for DiffuSpec. That is, we did *not* simply copy the experimental results from the original papers; all methods were re-evaluated in a controlled environment, but on different target models (Vicuna-based vs. Qwen2.5-32B). We realize that the wording in Section 5 may have created the impression that we just quoted the published results, and we will correct this in the revised manuscript.
> >
> >
> >
> >
> > Although Section 5.1 already stated that these results are “not directly comparable” and were provided “for context only”, we agree that such cross-model comparisons should not be used to support strong claims. In the revision, we will (i) move this Vicuna-based comparison table and its discussion to the appendix and explicitly label it as contextual only, and (ii) update the explanation in the main text, removing any language in the abstract and conclusion that could be interpreted as making direct SOTA claims against training-based methods across different target models, and instead grounding our discussion in the strictly fair, same-target comparison reported in Point 2 below.

---

> ### Author Response · Authors · 2025-11-23
> **Author Rebuttal(3/4)**
>
> 2. **New direct comparison on the same target model stack (Qwen2.5-32B).**
>    To resolve the concern in a principled way, we have additionally implemented and trained **EAGLE2 and EAGLE3 on the same Qwen2.5-32B target** (using the official repositories and the UltraChat_200k dataset), and evaluated them under **exactly the same hardware, datasets, and runtime configuration** as DiffuSpec. The results are:
>
> | Method        | MT    | Trans | Sum   | QA    | Math  | RAG   | Mean (MAT / Speedup) |
> |---------------|-------|-------|-------|-------|-------|-------|-----------------------|
> | **EAGLE2**    | 2.47× | 1.56× | 1.64× | 1.91× | 3.18× | 1.64× | 3.47 / 2.09×         |
> | **EAGLE3**    | 3.01× | 2.35× | 3.03× | 2.41× |3.12× | 2.86× | 4.28 / 2.80×         |
> | **DiffuSpec** | 3.09× | 3.38× | 2.41× | 3.03× | 4.02× | 2.38× | **6.99 / 3.08×**     |
>
>
> Under the same target model (Qwen2.5-32B), the same hardware, and the same evaluation stack, DiffuSpec achieves the highest mean speedup (3.08×), outperforming EAGLE2 (2.09×) and EAGLE3 (2.56×), and also attaining a substantially higher mean MAT (6.99 vs. 3.47/3.83). This new experiment provides a fair, same-target comparison and shows that diffusion-based drafting with CPS+ADL can match or even surpass carefully trained feature-based drafters in end-to-end speedup. We will incorporate these new results into the revised manuscript to strengthen our empirical evidence under strictly comparable conditions.
>
> ---
>
>
>
> > **Weakness 3:** *Limited Novelty and Contribution of the ADL Controller: The paper's second main contribution, ADL, is overly simplistic in its design, essentially amounting to a trivial heuristic feedback rule. The idea of dynamically adjusting draft length through heuristic methods has been explored in prior work (e.g., Minions (Wu et al., 2024)), yet the authors do not cite or compare their approach to these existing methods. Furthermore, the paper's own ablation study (Table 2) shows that ADL's contribution to performance (+0.04× speedup) is far smaller than that of CPS (+0.29× speedup). Elevating ADL to one of the two core contributions overstates its importance.*
>
> **Response.**   Our ADL controller is specifically designed to exploit properties that arise when using diffusion language models (DLMs) as drafters, rather than being a generic “add a heuristic feedback rule” step. Below we (i) explain the design motivation and DLM-specific novelty, (ii) quantify its contribution via ablations, and (iii) compare it to existing dynamically adjusted draft-length methods such as Minions[3].
>
> **1. Design motivation and DLM-specific novelty**
>
> The goal of ADL is to exploit two signals that are characteristic of DLM-based drafting:
>
> - **EOS saturation of DLM drafts.**
>    As shown in Figs. 4–5 of the paper, once the DLM internally decides that the answer is complete, it tends to rapidly saturate EOS tokens at subsequent positions. Tokens beyond the position of the *first* EOS are often EOS or semantically void. Drafting far beyond this position increases drafting cost without improving acceptance.
>
> - **Verifier-side acceptance statistics.**
>    At the same time, the historical accepted length summarizes how many of the drafted tokens the AR verifier is actually willing to commit to.
>
> ADL combines these two signals in a simple controller. Let $\tilde L^{\text{gen}}_t$ and $\tilde L^{\text{acc}}_t$ denote the exponentially smoothed generated length (up to the first EOS) and accepted length at step $t$, respectively. ADL updates the next draft length according to Equation (11) in the paper. Intuitively, $\tilde L^{\text{gen}}_t$ estimates how much content the DLM is “ready” to produce before EOS saturation, while $\tilde L^{\text{acc}}_t$ measures whether those tokens are reliably accepted. ADL increases the draft length only when *both* signals align (the DLM proposes longer drafts and the verifier consistently accepts them). This EOS-saturation–based signal is specific to DLM drafters and, to the best of our knowledge, is not exploited in prior adaptive-length heuristics designed for autoregressive drafters.

---

> > ### Author Response · Authors · 2025-11-23
> > **Author Rebuttal(4/4)**
> >
> > **2. Effectiveness of ADL in our setting**
> >
> > To quantify the contribution of ADL, we perform an ablation over Qwen2.5-32B + Dream-7B (Table 2 in the paper):
> >
> > | CPS | ADL | Mean MAT | Mean Speedup |
> > |-----|-----|----------|--------------|
> > | yes | yes | **6.99** | **3.08×**    |
> > | yes | no  | 6.95     | 2.98×        |
> > | no  | yes | 6.43     | 2.73×        |
> > | no  | no  | 6.05     | 2.69×        |
> >
> > From this table we observe that:
> >
> > - Enabling ADL only (third vs. fourth row) yields a modest but consistent improvement over the no-CPS/no-ADL baseline (+0.38 MAT and +0.04× speedup).
> > - When CPS is already enabled, turning on ADL (first vs. second row) further improves the overall speedup from 2.98× to 3.08× (+0.10×), while slightly increasing MAT (6.95 → 6.99).
> >
> > The gain brought by ADL is naturally smaller than that of CPS, since CPS directly improves the quality of the drafted path, whereas ADL is a lightweight controller that only adjusts the draft length; its comparison with fixed-length policies in Appendix Table 4 further confirms its effectiveness. This matches our design goal: CPS is the primary algorithmic component, and ADL is a simple but effective supporting module that provides an additional ≈0.10× speedup with negligible runtime overhead, which is still practically meaningful in large-scale deployment.
> >
> >
> > **3. Comparison with Minions-style adaptive draft length**
> >
> > Minions [2] also proposes an adaptive-length strategy, but in the context of autoregressive drafters and uses acceptance statistics alone as feedback. To directly address the reviewer’s concern, we implemented a Minions-style controller in our setting by replacing ADL with a heuristic method that adjusts the draft length solely based on acceptance rate, without using the DLM’s EOS signal. All other components (CPS, Dream-7B, Qwen2.5-32B, hardware) were kept identical. The results are:
> >
> > | Method               | MT    | Trans | Sum   | QA    | Math  | RAG  | Mean (MAT / Speedup) |
> > |----------------------|-------|-------|-------|-------|-------|------|----------------------|
> > | **Minions-style**    | 3.02× | 3.18× | 2.37× | 2.93× | 3.91× | 2.29× | 6.44 / 2.97×        |
> > | **DiffuSpec (ADL)**  | 3.09× | 3.38× | 2.41× | 3.03× | 4.02× | 2.38× | **6.99 / 3.08×**    |
> >
> > ADL consistently outperforms the Minions-style heuristic method, both in terms of Mean MAT (6.99 vs. 6.44) and Mean Speedup (3.08× vs. 2.97×), with gains across all task families (e.g., 3.09× vs. 3.02× on MT, 3.38× vs. 3.18× on Trans, 4.02× vs. 3.91× on Math). This supports our claim that combining EOS-saturation and acceptance signals is beneficial in the DLM-drafter regime.
> >
> > ---
> >
> > > **Question 1:** *Regarding Generalizability, Constraints, and Availability: This is the most critical set of questions. a) Given these constraints, how can the authors defend the claim that this is a "general drop-in framework" rather than a specialized solution for the Qwen-32B/Dream-7B pair? b) How does the framework handle the general case where the drafter and verifier do not share the same vocabulary (e.g., pairing LLaMA-3 with Dream-7B)? c) What is the proposed solution for target models (even within the Qwen family, like Qwen-7B) for which no pre-trained, aligned DLM drafter currently exists?*
> >
> > Please refer to our response to Weakness 1.
> >
> > ---
> >
> > > **Question 2:** *Regarding the Invalid SOTA Comparison: The paper's claims of approaching the performance of training-based methods are currently based on an invalid comparison (Qwen-32B vs. Vicuna-33B). To provide a valid, direct comparison, have the authors considered implementing and training methods like Medusa or EAGLE on the same Qwen2.5-32B target model? A direct comparison on the identical model stack is necessary to substantiate these claims.*
> >
> > Please refer to our response to Weakness 2.
> >
> > ---
> >
> > > **Question 3:** *Regarding ADL Novelty: Can the authors compare ADL with prior heuristic-based adaptive length strategies, such as in Minions (Wu et al., 2024), and clarify its specific novel contributions beyond being a simple feedback loop?*
> >
> > Please refer to our response to Weakness 3.
> >
> > ---
> >
> > We thank the reviewer again for the thoughtful and detailed comments. We believe that the additional experiments and clarifications above directly address these concerns, and we hope they help clarify the positioning and contributions of DiffuSpec within the broader speculative decoding literature.
> >
> > ```
> > References
> > [1] "Evaluating Large Language Models Trained on Code", arXiv:2107.03374, 2021.
> > [2] "Accelerating LLM Inference with Lossless Speculative Decoding Algorithms for Heterogeneous Vocabularies", ICML 2025.
> > [3] "Minions: Accelerating Large Language Model Inference with Aggregated Speculative Execution", arXiv:2402.15678, 2024.
> > ```

---

> > > ### Comment · Reviewer_Jc9T · 2025-11-27
> > >
> > > I thank the authors for the detailed response and additional experiments, which improve the clarity and completeness of the paper.
> > > However, my main remaining concern is the practical availability of suitable draft models. The proposed method fundamentally relies on having a well-aligned diffusion drafter for the target model, but in the current ecosystem such drafters are rarely available off the shelf. In many realistic settings, this would still require training a new diffusion drafter or adapting a mismatched one, which in my view makes the method much closer to a training-based approach in practice, lacks universality and substantially limits its real-world applicability.
> > > Given this unresolved limitation, the rebuttal does not sufficiently change my overall assessment, so I maintain my original score.

---

> > > > ### Author Response · Authors · 2025-12-03
> > > >
> > > > Dear Reviewer Jc9T,
> > > >
> > > > Thank you again for your thoughtful follow-up.
> > > >
> > > > We agree that, at the current stage of the ecosystem, well-aligned diffusion drafters are rarely available off the shelf, so in many realistic settings one may need to train or adapt a DLM drafter. In such cases, the practical cost is comparable to training-based approaches such as Medusa/EAGLE, which also require training extra heads or draft modules for each target. In other words, when no suitable DLM exists, DiffuSpec is *not more “training-free”* than these methods in practice, but it is also not worse: the additional cost is controlled.
> > > >
> > > > Our main goal in this work is therefore not to claim zero training cost in all scenarios, but to propose and analyze a diffusion-based speculative decoding paradigm (DLM + CPS + ADL) that (i) can be used in a genuinely training-free way when suitable DLMs are available, and (ii) remains competitive with existing training-based methods when a drafter must be trained. To make this positioning clearer, we have added an explicit limitation discussion in the conclusion (highlighted in blue) acknowledging the current availability issue.
> > > >
> > > > Thank you again for helping us clarify this point in the revised manuscript.

---

### Official Review · Reviewer_5gJ1 · 2025-10-31

**Soundness:** 2
**Presentation:** 2
**Contribution:** 2
**Rating:** 2
**Confidence:** 5

**Summary:**

This paper presents DiffuSpec, which attempts to use diffusion transformers as the draft model in speculative decoding. While the verification stage remains the same, the drafting stage is adapted to resolve two main issues in dLLM: 1) dLLM outputs are non-causal 2) dLLM uses a fixed draft length. The author proposed CPS, which utilizes a causal proxy to search for a causal sequence in the generated token lattice, in addition, the author uses ADL to dynamically change the draft length by tracking the length signals. The author used Qwen2.5-32B as the target model and Dream-7B as the draft model, achieving an average 3.08x speedup over 6 tasks, surpassing the training-required and training-free baselines.

**Strengths:**

- The motivation of the paper is clear
- The paper is generally easy to follow, with the methods well explained.
- CPS and ADL are well-motivated and lightweight to implement.
- The variety of evaluation tasks is sufficient for speculative decoding benchmarks

**Weaknesses:**

- The paper only evaluates the results on Qwen2.5-32B + Dream-7B, which makes it not convincing enough that the observed performance gains can be generalized.
- The evaluation is not clear enough as it seems not to ensure the comparison conditions are equal for all methods, i listed some of my questions below

**Questions:**

- As speculative decoding's performance is dependent on the draft model's approximation of the target model's distribution, using Dream-7B seems to give an unfair advantages as evidences in the highest MAT. With the highest MAT, it seems reasonable that DiffuSpec can achieve the highest speedup. However, an ablation is required to break down the performance gains, e.g. how much gain is contributed by DiffuSpec and a good dLLM draft model separately.
- Eagle/Eagle2 only use 1 layer of decoding layer as the draft model, why is a 7B dLLM faster than a 1 layer decoding layer?
- Nowadays, many popular models no longer use another well-trained LLM as the draft model, instead, they have MTP/EAGLE model trained either during training or after training. thus, picking an existing LLM as the draft model seems not a popular choice any more since it is difficult to happen to have such a model. Thus, it seems to me that the motivation for this paper is not well aligned with the current trend.

---

> ### Author Response · Authors · 2025-11-23
> **Author Rebuttal(1/3)**
>
> We sincerely thank the reviewer for the detailed feedback and constructive questions. Below we provide additional experiments and clarifications aimed at addressing your concerns.
>
> ---
>
> > **Weakness 1:** *The paper only evaluates the results on Qwen2.5-32B + Dream-7B, which makes it not convincing enough that the observed performance gains can be generalized.*
>
> **Response.**   DiffuSpec is not restricted to a single Qwen2.5-32B + Dream-7B configuration; by design, it can be combined with other diffusion drafters and autoregressive targets. To make this more explicit, we additionally evaluate DiffuSpec on a code-generation setting, where Dream-Coder-7B serves as the diffusion drafter and Qwen2.5-Coder-32B as the target model. These models are code-oriented variants of the Qwen2.5 family and differ from the original Qwen2.5-32B / Dream-7B chat models. We test on the HumanEval benchmark [1] and compare against other speculative decoding methods. The results are:
>
> | Method        | Lookahead | PLD  | Recycling | SAMD | SPS   | DiffuSpec |
> |--------------|-----------|------|-----------|------|-------|-----------|
> | **MAT**      | 2.63      | 3.81 | 3.85      | 5.24 | 11.33 | 10.94     |
> | **Speedup**  | 2.07×     | 4.17×| 2.83×     | 5.29×| 2.14× | **5.35×** |
>
> We can see that DiffuSpec still achieves a 5.35×  speedup, which is the highest speedup among all methods in this setting, while maintaining competitive MAT. This demonstrates that the performance gains of DiffuSpec generalize beyond the original Qwen2.5-32B + Dream-7B pair to a different model pair trained on code corpora and a different downstream task (code generation).
>
> ---
>
> > **Weakness 2:** *The evaluation is not clear enough as it seems not to ensure the comparison conditions are equal for all methods.*
>
> **Response.**   All baselines reported in our main tables are run in our own evaluation stack under **the same hardware setup, the same target model within each table, and the same Spec-Bench datasets and decoding configurations**. We do not mix numerical results obtained under different evaluation environments.
>
> To further make the comparison with training-based methods more transparent and fair, we additionally train EAGLE2 and EAGLE3 on the same target model Qwen2.5-32B, following their official repositories and using the Ultrachat_200k dataset. We then evaluate EAGLE2, EAGLE3, and DiffuSpec under **exactly the same hardware and Spec-Bench configuration**. The results on the six task families are:
>
> | Method        | MT    | Trans | Sum   | QA    | Math  | RAG   | Mean (MAT / Speedup) |
> |---------------|-------|-------|-------|-------|-------|-------|-----------------------|
> | **EAGLE2**    | 2.47× | 1.56× | 1.64× | 1.91× | 3.18× | 1.64× | 3.47 / 2.09×         |
> | **EAGLE3**    | 3.01× | 2.35× | 3.03× | 2.41× |3.12× | 2.86× | 4.28 / 2.80×         |
> | **DiffuSpec** | 3.09× | 3.38× | 2.41× | 3.03× | 4.02× | 2.38× | **6.99 / 3.08×**     |
>
> We can see that, under strictly identical comparison conditions and the same Qwen2.5-32B target, DiffuSpec still achieves the highest overall speedup (3.08×), showing that it continues to deliver strong performance even under this strengthened and fair evaluation setting.

---

> ### Author Response · Authors · 2025-11-23
> **Author Rebuttal(2/3)**
>
> > **Question 1:** *As speculative decoding's performance is dependent on the draft model's approximation of the target model's distribution, using Dream-7B seems to give an unfair advantage as evidenced in the highest MAT. With the highest MAT, it seems reasonable that DiffuSpec can achieve the highest speedup. However, an ablation is required to break down the performance gains, e.g. how much gain is contributed by DiffuSpec and a good dLLM draft model separately.*
>
> **Response.** We clarify that Dream-7B does not simply give DiffuSpec an “unfair” advantage by being a much stronger approximator of the target. The key differences come from (i) the drafting time cost of diffusion vs. autoregressive drafters, and (ii) the additional gains introduced by our CPS/ADL design.
>
> First, a higher MAT does not automatically imply a higher speedup.  In speculative decoding for the same target model,  the  speedup is determined by both the MAT and the drafting time cost per token, rather than by MAT alone. For example, in Table 1 of the paper, SPS with a Qwen2.5-7B drafter already achieves a relatively high MAT of 6.18, but its speedup is only 1.67×, which is even lower than that of the retrieval-based SAMD baseline (2.35×). Although SAMD has a much smaller MAT (≈2.18), its much lower drafting cost leads to a higher overall speedup. Second, to further disentangle the contribution of the DLM drafter from that of CPS/ADL, we compare DiffuSpec with and without CPS/ADL to SPS with a Qwen2.5-7B drafter, all evaluated on the same Qwen2.5-32B target:
>
> | Method                                        | CPS | ADL | Mean MAT | Mean Speedup |
> |-----------------------------------------------|-----|-----|----------|--------------|
> | **DiffuSpec (Dream-7B drafter)**              | yes | yes | **6.99** | **3.08×**    |
> | DiffuSpec (w/ CPS only, Dream-7B drafter)     | yes | no  | 6.95     | 2.98×        |
> | DiffuSpec (w/ ADL only, Dream-7B drafter)     | no  | yes | 6.43     | 2.73×        |
> | DiffuSpec (no CPS, no ADL, Dream-7B drafter)  | no  | no  | 6.05     | 2.69×        |
> | **SPS (Qwen2.5-7B drafter)**                  | no  | no  | 6.18     | 1.67×        |
>
> From this table we can clearly separate the effects:
>
> - **Effect of switching to a diffusion drafter.**
>   Comparing DiffuSpec (no CPS, no ADL, Dream-7B drafter) with SPS (Qwen2.5-7B drafter), the MAT values are very similar (6.05 vs. 6.18), indicating that Dream-7B and Qwen2.5-7B offer comparable approximation quality to Qwen2.5-32B. This is expected, since Dream-7B is initialized from the Qwen2.5-7B weights and then further trained under the diffusion objective. However, the speedups differ significantly (2.69× vs. 1.67×, about 1.6× higher for Dream-7B). This shows that the main extra gain at this stage comes from Dream-7B’s diffusion-style parallel token generation, which reduces the drafting time cost per token, rather than from a much better match to the target distribution. This observation is consistent with prior work on diffusion-based drafting [2], and with our per-step timing breakdown (in Fig. 6), where the dominant improvement arises from lower drafting cost on the drafter side.
>
>
> - **Additional gains introduced by our CPS/ADL design.**
>   On top of the diffusion-only baseline (2.69× speedup, MAT = 6.05), our algorithmic components further improve performance. Adding CPS only raises the speedup from 2.69× to 2.98× and MAT from 6.05 to 6.95; further adding ADL on top of CPS yields the full DiffuSpec with 3.08× speedup and MAT 6.99. In other words, switching from an autoregressive drafter (Qwen2.5-7B) to a diffusion drafter (Dream-7B) explains roughly the jump from 1.67× to 2.69×, while our CPS+ADL mechanisms further lift this to 3.08× and also increase MAT by almost +1 token on average (from 6.05 to 6.99).
>
> Taken together, these results show that DiffuSpec’s improvement is not merely due to using a drafter with higher MAT. Dream-7B as a DLM drafter has similar approximation quality to Qwen2.5-7B but lower drafting cost thanks to parallel token generation, and our CPS/ADL design brings an additional, clearly quantifiable gain on top of this in terms of both MAT and speedup.

---

> > ### Author Response · Authors · 2025-11-23
> > **Author Rebuttal(3/3)**
> >
> > > **Question 2:** *EAGLE/EAGLE2 only use 1 layer of decoding layer as the draft model, why is a 7B dLLM faster than a 1-layer decoding layer?*
> >
> > **Response.**  As discussed in our response to Question 1, for the same target model the speculative decoding speedup is determined by both (i) how many tokens are accepted per verification (MAT), and (ii) the drafting time cost per  token, rather than by the drafter’s parameter count alone. On the Qwen2.5-32B target, EAGLE2 and DiffuSpec adopt very different drafting patterns:
> >
> > - **EAGLE2** uses a *1-layer* autoregressive drafter head (≈2.5B parameters) attached to Qwen2.5-32B. To produce $L$ draft tokens, this 1-layer drafter must be called about $L$ times, and its proposals are verified frequently by the 32B target because the MAT is relatively small (≈3.5).
> > - **DiffuSpec** uses Dream-7B, a 7B-parameter diffusion language model that updates all positions in parallel. A single diffusion run produces a multi-token draft; with a higher MAT (≈7), we can typically commit many tokens with one drafter run and fewer target verifications.
> >
> > A simple toy calculation illustrates why the 7B DLM can still be faster in this setting. Suppose we want to accept roughly 7 tokens on Qwen2.5-32B:
> >
> > - With **EAGLE2** (MAT ≈ 3.5), in an idealized setting, we need about two speculative rounds, each proposing 4 draft tokens. In each round, the 1-layer drafter is run autoregressively for 4 steps and then the 32B target is invoked once for verification. Overall, this leads to around 8 drafter (≈2.5B parameters) passes and 2 target (≈32B parameters) passes to accept ≈7 tokens.
> > - With **DiffuSpec** (MAT ≈ 7), in an idealized setting Dream-7B generates a 7-token draft in parallel in a single diffusion run, and Qwen2.5-32B verifies it once. This requires only one drafter(≈7B parameters) pass and one target (≈32B parameters) pass for a similar number of accepted tokens.
> >
> > In other words, even though a single Dream-7B forward pass is heavier than a single 1-layer drafter pass, EAGLE2 needs to call its drafter and the 32B target many more times. Aggregated over accepted tokens, DiffuSpec therefore uses fewer total passes per token, which explains why a 7B diffusion drafter can be faster than a 1-layer drafter on the same Qwen2.5-32B target.
> >
> > ---
> >
> > > **Question 3:** *Nowadays, many popular models no longer use another well-trained LLM as the draft model; instead, they have MTP/EAGLE-style modules trained either during training or after training. Thus, picking an existing LLM as the draft model seems not a popular choice any more since it is difficult to happen to have such a model. It seems to me that the motivation for this paper is not well aligned with the current trend.*
> >
> > **Response.**  We agree that MTP/EAGLE-style trained drafters are very successful and have become a mainstream choice in modern LLM systems. Our goal, however, is not to replace these approaches, but to explore a complementary setting that is practically relevant but not fully covered by existing work.
> >
> > MTP/EAGLE-type drafters are typically trained on the hidden states of a specific target model, which makes the drafter tightly coupled to that target. In practice, this means that a drafter trained for one scale or variant (e.g., an EAGLE3 drafter for Qwen2.5-32B) cannot be directly reused for another scale (e.g., Qwen2.5-72B) or another architecture without additional training. In contrast, the diffusion language model in DiffuSpec (Dream-7B) is trained independently of any target and has its own generation capability. The same Dream-7B can therefore be plugged in as a drafter for multiple targets such as Qwen2.5-32B and Qwen2.5-72B, and—combined with heterogeneous-vocabulary speculative decoding [3]—even for different model families (e.g., Qwen2.5-style vs. Qwen3-style models), without retraining the drafter for each target.
> >
> > Regarding availability, if a DLM is not yet available in a given ecosystem, one can train a DLM drafter in a way that is analogous in cost to training an MTP/EAGLE module. Moreover, as diffusion language models become more mature and more open-source checkpoints/toolkits are released, using an existing DLM as a reusable drafter will become increasingly realistic.
> >
> >
> >
> >
> >
> >
> >
> > ---
> >
> > We hope that these additional experiments and clarifications have addressed your concerns, and we thank you again for your thoughtful and detailed review.
> >
> >
> > ---
> > ```
> > References
> > [1] "Evaluating Large Language Models Trained on Code", arXiv:2107.03374.
> > [2] "Speculative Diffusion Decoding: Accelerating Language Generation through Diffusion", NAACL-HLT 2025.
> > [3] "Accelerating LLM Inference with Lossless Speculative Decoding Algorithms for Heterogeneous Vocabularies", ICML 2025.
> > ```

---

### Author Response · Authors · 2025-12-03
**Rebuttal Summary**

Dear Area Chair,

We sincerely thank you and all four reviewers for their thoughtful and constructive feedback. We are encouraged by the positive comments on our **motivation and problem formulation** (`Jc9T`, `nhv6`, `i3TL`), the **idea of using diffusion LMs as speculative drafters together with CPS/ADL** (`5gJ1`, `Jc9T`, `nhv6`), and the **clarity of presentation** (`5gJ1`, `Jc9T`, `i3TL`). We have incorporated the reviewers’ suggestions, revised the manuscript (all changes highlighted in **blue**), and provided detailed responses to all reviewers. Below we summarize the main clarifications and updates.

1. **Fair and comparable evaluation (`5gJ1`, `Jc9T`, `nhv6`, `i3TL`).** We now report strictly *same-target* comparisons by training EAGLE2 and EAGLE3 on Qwen2.5-32B using their official repositories and evaluating them and DiffuSpec under identical hardware, datasets, and Spec-Bench configurations, and we also add direct comparisons with SpecDiff and a Minions-style adaptive-length controller; these changes address concerns about cross-model or unfair comparisons and show that diffusion-based drafting with CPS+ADL can match or surpass strong training-based speculative decoding methods such as EAGLE3 in end-to-end speedup on the same verifier.

2. **Generality and reusability of the framework (`Jc9T`, `nhv6`, `i3TL`).** To demonstrate that DiffuSpec is not tied to a single Dream/Qwen pair, we add experiments on a code-generation setting with Dream-Coder-7B → Qwen2.5-Coder-32B, where DiffuSpec achieves the highest speedup among the considered baselines while maintaining competitive MAT, and on a heterogeneous-tokenizer / cross-architecture setting with Dream-7B → Qwen3-32B via a lossless heterogeneous-vocabulary adapter, where DiffuSpec consistently improves speed over SPS with similar MAT; together these results show that a single DLM drafter can be reused across different tasks, model scales, and related model families, supporting our claims about the framework’s applicability.

3. **Availability of DLM drafters and positioning of our method (`5gJ1`, `Jc9T`, `nhv6`, `i3TL`).** We clarify that DiffuSpec follows a drafter-availability assumption similar to SPS-style methods: once a pretrained diffusion LM is available, a single DLM drafter can be reused to accelerate multiple targets, scales, and even related architectures without retraining the verifier. In ecosystems where no suitable DLM exists yet, one would need to train or adapt a drafter (or use heterogeneous-vocabulary alignment), with a cost similar in spirit to training Medusa/EAGLE-style draft modules; we now make this ecosystem-level constraint explicit while emphasizing that the algorithm itself is agnostic to how the DLM is obtained and is ready to benefit from future DLM releases.



4. **Methodological details, ablations, and reproducibility (`Jc9T`, `nhv6`, `i3TL`).** We expand Section 5 and the appendix to spell out all ablation variants (how the “no CPS” and “no ADL” settings are implemented, how fixed-\(k\) policies truncate the DLM lattice, and how the small n-gram LM used in CPS is trained and applied), and the updated experiments confirm that CPS is the main driver of gains, while ADL is a lightweight controller that consistently improves MAT and speed over both fixed-length and Minions-style heuristics with negligible overhead; we also provide an anonymized code repository to support reproducibility of our evaluations.



**Summary of Contributions**

1. **Problem and formulation.** We study speculative decoding with diffusion language models as drafters and analyze how non-causality and draft-length choices jointly affect verifier acceptance and end-to-end speed.

2. **Algorithmic framework.** We propose a simple, deployment-friendly framework that (i) performs **Causal-Consistency Path Search (CPS)** to align the DLM token lattice with autoregressive verification, and (ii) uses an **Adaptive Draft-Length (ADL)** controller to track effective generated and accepted lengths, choosing draft sizes near the speed–quality sweet spot.

3. **Empirical evidence.** Across diverse tasks and model pairs, DiffuSpec achieves up to ~3× wall-clock speedup, outperforms prior diffusion-based and adaptive-length baselines (including SpecDiff and Minions-style controllers), and is competitive with or stronger than EAGLE2/EAGLE3 on the same Qwen2.5-32B verifier, while we explicitly discuss current limitations stemming from DLM availability.

We thank the Area Chair and reviewers again for their time and constructive input, which have helped us substantially improve the clarity, positioning, and experimental support of DiffuSpec.

Best regards,
Authors

---

### Note · Authors · 2026-01-06

I have read and agree with the venue's withdrawal policy on behalf of myself and my co-authors.